# Surfactant Protein-G in Wildtype and 3xTg-AD Mice: Localization in the Forebrain, Age-Dependent Hippocampal Dot-like Deposits and Brain Content

**DOI:** 10.3390/biom12010096

**Published:** 2022-01-07

**Authors:** Anton Meinicke, Wolfgang Härtig, Karsten Winter, Joana Puchta, Bianca Mages, Dominik Michalski, Alexander Emmer, Markus Otto, Karl-Titus Hoffmann, Willi Reimann, Matthias Krause, Stefan Schob

**Affiliations:** 1Paul Flechsig Institute for Brain Research, University of Leipzig, Liebigstr. 19, 04103 Leipzig, Germany; anton.meinicke@medizin.uni-leipzig.de (A.M.); wolfgang.haertig@medizin.uni-leipzig.de (W.H.); joana.puchta@gmail.com (J.P.); willi.reimann@medizin.uni-leipzig.de (W.R.); 2Institute of Neuroradiology, University Hospital Leipzig, Liebigstr. 20, 04103 Leipzig, Germany; Karl-Titus.Hoffmann@medizin.uni-leipzig.de; 3Institute of Anatomy, University of Leipzig, Liebigstr. 13, 04103 Leipzig, Germany; kwinter@rz.uni-leipzig.de (K.W.); bianca.mages@medizin.uni-leipzig.de (B.M.); 4Department of Neurology, University Hospital Leipzig, Liebigstr. 20, 04103 Leipzig, Germany; Dominik.michalski@medizin.uni-leipzig.de; 5Department of Neurology, University Hospital Halle, Ernst-Grube-Str. 40, 06120 Halle (Saale), Germany; Alexander.Emmer@uk-halle.de (A.E.); markus.otto@uk-halle.de (M.O.); 6Department of Neurosurgery, University Hospital Leipzig, Liebigstr. 20, 04103 Leipzig, Germany; m.krause@medizin.uni-leipzig.de; 7Department of Neuroradiology, Clinic and Policlinic of Radiology, University Hospital Halle, Ernst-Grube-Str. 40, 06120 Halle (Saale), Germany

**Keywords:** Alzheimer’s disease, SP-G, SFTA2, hippocampus, habenula, Reelin, 3xTg mouse, β-amyloid, hyperphosphorylated tau

## Abstract

The classic surfactant proteins (SPs) A, B, C, and D were discovered in the lungs, where they contribute to host defense and regulate the alveolar surface tension during breathing. Their additional importance for brain physiology was discovered decades later. SP-G, a novel amphiphilic SP, was then identified in the lungs and is mostly linked to inflammation. In the brain, it is also present and significantly elevated after hemorrhage in premature infants and in distinct conditions affecting the cerebrospinal fluid circulation of adults. However, current knowledge on SP-G-expression is limited to ependymal cells and some neurons in the subventricular and superficial cortex. Therefore, we primarily focused on the distribution of SP-G-immunoreactivity (ir) and its spatial relationships with components of the neurovascular unit in murine forebrains. Triple fluorescence labeling elucidated SP-G-co-expressing neurons in the habenula, infundibulum, and hypothalamus. Exploring whether SP-G might play a role in Alzheimer’s disease (AD), 3xTg-AD mice were investigated and displayed age-dependent hippocampal deposits of β-amyloid and hyperphosphorylated tau separately from clustered, SP-G-containing dots with additional Reelin-ir—which was used as established marker for disease progression in this specific context. Semi-quantification of those dots, together with immunoassay-based quantification of intra- and extracellular SP-G, revealed a significant elevation in old 3xTg mice when compared to age-matched wildtype animals. This suggests a role of SP-G for the pathophysiology of AD, but a confirmation with human samples is required.

## 1. Introduction

Surfactant proteins (SPs) are essential components of the pulmonary surface active agent ‘surfactant’, a lipoprotein complex lining the inner surface of the alveoli that most importantly decreases the surface tension of the alveolar fluid [1]. Surfactant functionality is required to prevent lung collapse during expiration and facilitates lung expansion during inspiration. Lack of surfactant in neonates causes fatal respiratory distress syndrome if left untreated and a decreased concentration of surfactant is also a critical factor for COVID-19 mortality in adults [2]. Four classic SPs have been described in the lungs—the SPs A, B, C, and D. SP-A and SP-D are hydrophilic collectin-type SPs predominantly involved in host defense and modulation of the immune response, whereas SP-B and SP-C are hydrophobic peptides that regulate the surface tension at the surface interface during the breathing cycles [3]. All of the classic SPs are inherent to the CNS [4,5]. However, SP-C seems to be particularly involved in regulating rheological properties of the cerebrospinal fluid (CSF) and its age-dependent decline is linked to increased white matter lesions [6,7]. Especially the latter fact is suggestive of SP-C’s involvement in brain waste clearance and indicates its potential protective function in context of neurodegeneration.

More recently, a novel surfactant protein—SP-G—has been introduced. It was first described in the context of an extensive bioinformatical investigation of cleavage sites of secretory proteins performed by Zhang et al. [8] as “Surfactant Associated 2” (SFTA2). Studies by Rausch et al. elucidated the properties of SP-G as an amphiphilic molecule which is most present in the lung, but can also be found elsewhere, e.g., in the cervix uteri, testis and brain [9]. Although the physiological roles of SFTA2 are still unknown, Rausch and co-workers were able to show a significant decline of SP-G in mice challenged by lipopolysaccharide instillation, indicating a possible role of SP-G in inflammatory processes [10]. Intraventricular hemorrhage in premature pediatric patients, as well as disorders of cerebrospinal fluid (CSF) circulation in adults, were associated with elevated levels of SP-G in the CNS, which, in combination with its amphiphilic properties, suggests the possibility of SP-G as being an important modulator of CSF homeostasis [11,12].Thus, SP-C and SP-G apparently share common features in this regard.

In the rat brain, SP-G-immunoreactivity (ir) was predominantly found within the ependymal lining of all ventricles and the choroid plexus, representing the morphological correlate of the blood–cerebrospinal fluid barrier. Furthermore, parenchymal SP-G showed a clear colocalization with endothelial CD31 and aquaporin 4 as a marker for astroglial endfeet, indicating its involvement in the blood–brain barrier [11]. Moreover, in neurons of the subventricular and superficial cortex, SP-G-ir was observed around and occasionally intermingled with Neuronal Nuclei (NeuN)-ir in the perineuronal zone. However, more detailed analyses focused on the distribution of SP-G in other mammals and its neurophysiological roles are lacking. Hence, these preliminary findings demand confirmation and should be extended comprehensively in the mouse as a second species. In detail, the cellular origins of SP-G and its compartmental distribution within the CNS require clarification. Thus, SP-G co-localization with astrocytes, marked by glial fibrillary acidic protein (GFAP) and S100β, oligodendrocytes immunoreactive for 2′,3′-cyclic-nucleotide 3′-phosphodiesterase (CNP), as well as microglia, detected by antibodies directed against the ionized calcium-binding adaptor molecule 1 (Iba-1), should be investigated. Furthermore, using the well-established approach first described by Hofstein et al. [13], the quantification of SP-G in the interstitium and in the intracellular space as well is important to better understand its physiological significance.

In addition, insight on SP-G’s regulation under pathological conditions, for example, in the context of neurodegeneration, is also lacking. As recently shown, SP-C is entangled in characteristic neuropathological changes in a murine model of Alzheimer’s disease (AD) [14], represented by Reelin/SP-C double-positive deposits that manifest in an age-dependent fashion [15]. According to Hoe et al., functional Reelin not only modulates synaptic plasticity but also distinctly limits hyperphosphorylation of tau and the formation of β-amyloid [16]. Contrarily, with increasing age, the deposition of non-functional Reelin precedes and enhances extracellular accumulation of β-amyloid in association with synaptic dysfunction [17,18]. Considering the yet known functional overlap between SP-C and SP-G, this prompts the question of whether SP-G expression patterns are also changed in this context, and more specifically, if they correlate with the neuropathological hallmarks in AD. As a consequence, determination of possibly age-dependent alterations of SP-G expression, predominantly in the hippocampus, combined with the detection of the neuropathological hallmarks of AD, e.g., tau hyperphosphorylation (for reviews see [19,20]) and β-amyloidosis (for reviews see [21,22,23]), are warranted. 

## 2. Materials and Methods

### 2.1. Animal Handling

The conducted animal experiments were carried out according to the European Union Directive 2010/63/EU. They had been approved by the local authority (Regierungs-präsidium Leipzig; reference number T12/16). 

### 2.2. Brain Tissue Preparation

Histochemical analyses based on 30 WT and 3xTg mice each with n = 6 per age group (3, 6, 12, 16, and 24 months). All animals were transcardially perfused, first with saline and then with 4% phosphate-buffered paraformaldehyde. Next, brains were removed from the skulls and post-fixed in the same fixative overnight. Subsequently, they were equilibrated with 30% phosphate-buffered sucrose. Forebrains were then sectioned with a freezing microtome (Leica SM 2000R, Leica Biosystems, Wetzlar, Germany) resulting in 10 series of 30 µm-thick coronal sections. They were stored at 4 °C in sealed glass vials containing 0.1 M Tris-buffered saline, pH 7.4 (TBS) with sodium azide as an additive.

For biochemical analyses, we used at least 5 whole brains from 3, 6, 12 and 16 month-old WT and age-matched 3xTg mice. Sacrificed animals were perfused with saline. Thereafter, all brains were removed from the skulls and divided into two hemispheres. Their different processing is described below under 2.5. (Biochemical analyses).

### 2.3. Fluorescence Labeling

All staining experiments were started by extensively rinsing the free-floating mouse forebrain sections with TBS. Next, to block its non-specific binding sites, the tissue was treated with 5% normal donkey serum in TBS containing 0.3% Triton X-100 (NDS-TBS-T) for 1 h.

SP-G-immunoreactivity (ir) was routinely detected with affinity-purified rabbit antibodies raised against the recombinant SP-G, as described previously [11].

For single staining and all multiple labeling in this study, rabbit-anti-SP-G (Hölzel, Cologne, Germany as supplier for Cloud-Clone, Katy, TX, USA; product number PAD755Hu01; 1:300 = 10 µg/mL NDS-TBS-T) was incubated with the sections for 20 h. After three rinses with TBS for 10 min each, the incubation with carbocyanine (Cy3)-conjugated donkey-anti-rabbit IgG then followed— all fluorochromated immunoreagents in this study were obtained from Dianova, Hamburg, Germany (as supplier for Jackson ImmunoResearch, West Grove, PA, USA) and used at 20 µg/mL in TBS, containing 2% bovine serum albumin (TBS-BSA) for 1 h.

For a first set of triple fluorescence staining, mixtures of anti-SP-G (in NDS-TBS-T) and the endothelial-binding biotinylated *Lycopersicon esculentum* (tomato) agglutinin (LEA; Vector Labs, Burlingame, CA, USA; 20 µg/mL) were completed by one of the following guinea pig antibodies (all from Synaptic Systems, Göttingen, Germany), used for detecting NVU components: NeuN (1:200), CNP (1:200), Iba (1:100), S100β (1:200), AQP4 (1:200), and GFAP (1:200). Sections were reacted with one of these mixtures for 20 h, rinsed with TBS and the markers were visualized by incubating them with a cocktail of Cy3-donkey-anti-rabbit IgG, Cy5-streptavidin, and Cy2-donkey-anti-guinea pig IgG for 1 h.

Next, sections from 16 month-old 3xTg-AD mice were incubated overnight with mixtures of rabbit-anti-SP-G (1:300 in NDS-TBS-T) and guinea pig-anti-GFAP (1:200), additionally containing biotinylated mouse antibodies directed against either Aβ_18–23_ (4G8; Biolegend, San Diego, CA, USA; 1:100) or phospho-tau_202+205_ (AT8; ThermoFisher Scientific, Schwerte, Germany; 1:50). The sections were then rinsed with TBS, followed by their concomitant incubation with Cy3-anti-rabbit IgG, Cy5-donkey-anti-guinea pig IgG, and Cy2-streptavidin for 1 h.

For subsequent semiquantitative analyses from all mice, 6 to 8 frontal sections comprising rostral and caudal planes of the hippocampus were primarily incubated with a mixture of rabbit-anti-SP-G (in NDS-TBS-T), mouse-anti-Reelin (Merck Millipore, Billerica, MA, USA, 1:100), and biotinylated *Solanum tuberosum* (potato) lectin (STL; Vector; 20 µg/mL) for 20 h. Thereafter, the sections were washed with TBS, followed by incubation with a cocktail of Cy3-donkey-anti-rabbit IgG, Cy2-donkey-anti-mouse IgG, and Cy5-streptavidin for 1 h.

In control experiments, primary antibodies or biotinylated lectins were omitted, which led to the expected absence of cellular labeling. Further controls for the detected age-dependent AD-like alterations (i) comprised the switch of fluorophores Cy2 and Cy5 visualizing the AD markers Aβ or phospho-tau_202+205_ and astroglial GFAP, and (ii) included the staining of age-matched, e.g., 16 month-old WT mice, as well as 3 month-old 3xTg-AD and WT mice, which were previously shown to be devoid of AD-like neuropathological alterations.

Finally, all sections were extensively rinsed with TBS and shortly placed in distilled water, mounted onto slides, air-dried, and coverslipped with Entellan in toluene (Merck, Darmstadt, Germany).

### 2.4. Microscopy and Image Processing

Fluorescently labeled brain sections were primarily screened with an Axioplan fluorescence microscope (Zeiss, Germany). Subsequent imaging of triple-stained sections was performed on a Biorevo BZ-9000 microscope (Keyence, Neu-Isenburg, Germany). A confocal laser-scanning microscope (LSM 880; Carl Zeiss AG, Jena, Germany) was used for the detailed imaging of selected sections. Panels of micrographs were generated by means of Microsoft PowerPoint (version 2015; Microsoft Corp., Redmond, WA, USA). Thereby, the brightness and contrast of micrographs were slightly adjusted in a few cases, but the deletion or creation of signals was circumvented.

For the semi-quantification of hippocampal dots, 4–6 sections per mouse, covering rostral, as well as caudal parts of the hippocampus, were chosen. The immunolabeled sections were run through a digital slide scanner (Panoramic Scan II, 3D HISTECH Ltd., Budapest, Hungary) to acquire digital images. A delineation of the hippocampal region was performed on the slide scanner data sets (Case Viewer, Version 2.3.0.99276, 3D HISTECH Ltd.) and exported as raster images (pixel dimensions 0.325 µm).

Images were imported into Mathematica (Version 12.0, Wolfram Research, Inc., Champaign, IL, USA) to perform image analysis. Therefore, they were divided into separate color channels of which the red channel was inverted followed by its subtraction from the green image channel. Next, Kittler-Illingworth minimum error thresholding was performed [24]. The results of the segmentation were exported and the GNU Image Manipulation Program (Version 2.8, [25]) was used to eliminate falsely positive artifacts, e.g., tissue overlaps or intraluminal precipitates. After this correction, images were loaded into Mathematica to compute the total area of the hippocampal region, the area of the segmented dots, and, finally, to calculate the Dot-to-Tissue-Ratios (DTR).

### 2.5. Biochemical Analyses

To determine the intracellular, extracellular, and total concentration of SP-G and Reelin with enzyme linked immunosorbent assays (ELISAs), the brains of all 54 mice were cut in half by following the interhemispheric gap. An amount of 1 mL of phosphate-buffered saline (PBS, pH 7.4) was added to one hemisphere which was then mechanically homogenized. 500 µL of this homogenate was mixed thoroughly with 300 µL of Radioimmunoprecipitation Assay buffer (RIPA; [26]). The supernatant containing the combined intra- and extracellular protein fraction (total protein fraction TCF) was pipetted off and stored at 4 °C.

The second hemisphere was incubated in extraction buffer (0.32 M sucrose, 1 mM CaCl_2_, 10 U/mL Heparin in Hank’s Balanced Salt Solution (HBSS) [27] for 2 h at room temperature. Next, the hemisphere was removed and processed—as described for TCF above—to acquire the intracellular protein fraction (ICF). The extraction buffer containing the extracellular protein fraction (ECF) was mixed with 1.3 mL of ice-cold acetone and incubated at 4 °C overnight to precipitate the proteins. The next day it was centrifuged for 15 min (4 °C, 3500× *g*), and the supernatant was discarded, while the pellet was air-dried and resuspended in 1 mL of PBS.

For the measurement of the total protein concentration in all obtained fractions, a BCA (bicinchoninic acid) Protein Assay Kit and a Spectramax M5 (Molecular Devices, LLC. San Jose, CA, USA) photometer running SoftMax Pro 5 were used. The ELISAs were performed according to the manufacturer’s user guide (Cloud-Clone, Fernhurst, TX, USA) and measured at 450 nm wavelength.

### 2.6. Statistical Analysis

All statistical analyses were performed with IBM SPSS Statistics (Version 22, IBM, Armonk, NY, USA). Next, descriptive statistics were performed, and box plots were generated. Obtained data was tested for normal distribution by the Shapiro-Wilk-Test. The group comparisons were carried out with the Kruskal-Wallis-Test, the Mann-Whitney-U Test or via an Analysis of Variance (ANOVA). Bonferroni correction was performed before post hoc analysis. The Spearman’s Rank-Order Correlation or Pearson’s Coefficient were calculated to test for associations between the different groups of age and genotype. For all tests, the level of significance was set to *p* < 0.05. To adjust the *p*-value for multiple comparisons, the Dunn’s post hoc test was used.

## 3. Results

The results are presented in the following way: At first, the findings of the comprehensive co-localization studies—aiming to determine cellular sources of SP-G in the murine forebrain—are given. Thereafter, the identified extracellular SP-G-ir patterns, manifesting as Reelin-SP-G-positive dots in different brain areas, are provided. As the next step, the semi-quantification of SP-G-immunoreactive structures in 3xTg- and wildtype mice of different ages is demonstrated in a comparative fashion, followed by the results of the ELISA-based quantification of SP-G from whole hemispheres, distinguishing the extracellular interstitium from the intercellular space.

Screening experiments of mouse forebrains revealed SP-G-expressing cells in the habenula and hippocampus and strong SP-G-ir in the infundibulum.

Age-dependently occurring SP-G-immunoreactive dot-like structures that were distinct from the cellular staining were also found. They appeared to be frequently clustered in the hippocampus, but also to a lower extent in the piriform cortex. Hippocampal areas displaying the most SP-G-immunopositive dots were the stratum lacunosum-moleculare and the stratum radiatum of the CA1 region.

### 3.1. Neurons, SP-G and Vessels

The figures are presented in a way that reflects the investigative strategy. At first, the chemical neuroanatomy of SP-G, including the spatial and functional relationships between SP-G and well-established markers for neurons, oligodendroglia, microglia, and astroglia, is provided in differently aged animals of both genetic backgrounds. In the following images, representative scenes of neuropathologically significant but spatially distinct depositions—SP-G, Reelin, β-amyloid, and phospho-tau—in the respective areas of interest are shown, also for both genetic backgrounds.

The qualitative analysis of spatial relationships between SP-G, LEA-positive vessels, and NVU components in 3 and 16 month-old WT- and 3xTg mice started with the counterstaining of neurons by NeuN-ir (Figure 1, Figure 2 and Figure 3).

Neuronal SP-G-ir was displayed by the habenula from a young WT mouse in Figure 1A,A’’’, also showing a less strong marked dentate gyrus and a faintly labelled granular layer. Confocal laser-scanning clearly verified SP-G-ir in numerous neurons located in the habenula (Figure 1B–B’’).

Neuronal SP-G-ir was also exemplified by Figure 2, showing immunopositive somata in the hypothalamus of a 3 month-old 3xTg-AD mouse at a lower magnification (Figure 2A–A’’’).

Thereby, numerous double-stained neurons presented SP-G-ir (red) around neuronal nuclei. Some further neurons devoid of a visible nucleus appeared red. Again, confocal-laser scanning micrographs demonstrated neurons co-expressing SP-G and NeuN (Figure 2B–B’’).

At higher magnification of hippocampal tissue in Figure 3A, mossy fibers of the CA3 region from an old WT mouse exhibited a bright SP-G-staining. Notably, clustered dots, mostly in stratum lacunosum-moleculare and stratum oriens of the regions CA1 and CA2 (arrow in Figure 3A) exhibited an even stronger SP-G-ir.

Such deposits resemble those containing reelin [28] and SP-C [15] and the age-associated inclusions described by Jucker, Walker and co-workers [29,30].

The counterstaining of NeuN-ir was strongest in the granular layer with densely packed neuronal somata, while NeuN-immunopositive hippocampal neurons were apparently devoid of SP-G-ir.

SP-G-containing cells and long processes in zone I of the infundibulum from a 3 month-old WT mouse might represent Kolmer/epiplexus cells. Notably, the infundibulum contained only some less densely packed neurons with NeuN-ir (Figure 3B’).

Concomitant staining with the biotinylated tomato lectin LEA revealed evenly distributed endothelia in all presented regions (Figure 1, Figure 2 and Figure 3). Remarkably, in Figure 3B’’’ the infundibulum displayed strongly SP-G-labeled fibers stretching out from cell somata in zone I to the capillaries adjacent to zone IV, and somewhat weaker immunosignals of ependymal cells encircling the third ventricle with a turquoise appearance as an indicator for SP-G-ir and LEA-binding sites were in close vicinity.

### 3.2. Oligodendroglia, SP-G and Vessels

For the detection of spatial relationships between SP-G-immunopositive structures, LEA-binding vessels, and oligodendroglia, CNP-ir was counterstained. Triple-labeled forebrain regions are shown in Figure 4A–D and represent examples for different animal groups (for details see the related figure legends). SP-G-containing somata were again observed in the habenula and appeared strongest in its medial part, but were also visible in the lateral habenula, the intermediodorsal, lateral posterior, parafascicular nuclei, and the posterior complex of the thalamus (Figure 4A), as well as in the hypothalamus (Figure 4D). All these brain regions were devoid of SP-G-containing clustered dots, which became visible only in the hippocampal dentate gyrus, strata lacunosum-moleculare and radiatum of the CA1 and CA2 fields (Figure 4B), and at a higher magnification in the moleculare layer of the dentate gyrus (Figure 4C).

Generally, immunoreactivities for oligodendroglial CNP and SP-G were not co-localized but CNP-immunoreactive structures enwrapped hypothalamic cells containing the surfactant protein. The LEA counterstaining of vessels was heaviest around the ventricle and in the fissura hippocampalis but entirely separate from the immunolabeling.

### 3.3. Microglia, SP-G and Vessels

Subsequently, Figure 5 exemplifies the concomitant fluorescence labeling of SP-G, LEA-positive vessels and microglial Iba by images from the hippocampus of a 16 month-old WT mouse. Numerous SP-G-immunoreactive dots were observed in stratum lacunosum-moleculare of the CA1 field, the molecular layer of the dentate gyrus (Figure 5A), and at a higher magnification in the stratum lacunosum-moleculare of the CA2 region (Figure 5B). In parallel, Iba-immunoreactive microglia (Figure 5A’,B’) were found regularly distributed in the presented regions. Additional STL-staining (Figure 5A’’,B’’) revealed the vasculature. The overlay (Figure 5A’’’) elucidated several larger perpendicularly cut vessels adjoining to clustered SP-G-immunoreactive dots on the border between the strata lacunosum-moleculare and radiatum. In addition, the nearly general absence of co-occurring SP-G and Iba became obvious. Both overlays (Figure 5A’’’,B’’’) clearly show that the SP-G-immunopositive dots are devoid of Iba-ir, which was exclusively found in ramified microglia.

As an incidental finding, Iba-immunopositive cells, which also showed intracellular SPG immunoreactivity, were found within the choroid plexus in two sections from the same animal. Subsequently, another series of sections from a total of nine mice was prepared to further investigate this finding. The initial co-expression did not show up in any further sections but will nevertheless be briefly discussed at a later point (Data not shown).

### 3.4. Astroglial Markers, SP-G and Vessels

To complete the qualitative analyses of spatial relationships between SP-G-ir, LEA-binding sites, and further NVU components, additional multiple labeling comprised the immunodetection of astrocytic S100β (Figure 6A–A’’’,B), AQP4 (Figure 6C–C’’), and GFAP (Figure 7). In a 16 month-old WT mouse, clusters of SP-G-containing dots were found in the hippocampal stratum oriens and in the strata lacunosum-moleculare and radiatum (Figure 6B), but notably also in the piriform cortex, as clearly shown in Figure 6A.

Without exception, the counterstained, evenly distributed, S100β-immunoreactive astroglia and vascular LEA-binding sites were distinct from SP-G-ir in cells and dots (Figure 6A’–A’’’). On the contrary, the habenula of a young 3xTg-AD mouse (Figure 6C–C’’) displayed cellular SP-G-ir, while AQP4-immunolabeled vessels were predominantly located in the medial habenula close to the ventricle. The merge of staining patterns also indicated the close vicinity of AQP-4-ir and LEA-binding sites in vessels, with a turquoise appearance.

Next, triple fluorescence staining of SP-G-ir, LEA-binding sites and GFAP was first exemplified by the higher magnified hippocampal CA1 region from a 16 month-old 3xTg mouse (Figure 7A–A’’’).

Thereby, many clustered, SP-G-containing dots were found (Figure 7A) concomitantly with astrocytes, which appeared at least partially activated (Figure 7A’), and LEA-binding sites as to be expected in endothelia but in also structures resembling plaques (Figure 7A’’).

After merging of the staining patterns (Figure 7A’’’), reactive astrocytes (arrow) were observed around lectin-positive plaques (arrowheads). Next, the infundibulum from a 3 month-old WT mouse (Figure 7B–B’’’) displayed several parallelly oriented SP-G-immunoreactive fibers arising in zone I and stretching through zones II to IV up to the adjacent capillaries. The infundibulum was nearly devoid of visible GFAP-ir (Figure 7B’), but astroglial counterstaining was strong in the adjoining arcuate hypothalamic nucleus, whereas heavy LEA-staining (Figure 7B’’) was observed in the attached meninges and around the ventricle. At a higher magnification (Figure 7B’’’), the layer encircling the ventricle appeared predominantly purple, as a sign for co-occurring SP-G-ir (red) and LEA-binding sites (blue).

### 3.5. Clustered SP-G Dots, β-Amyloid Deposits, and Hyperphosphorylated Tau

In order to reveal spatial relationships between age-dependently occurring β-amyloid deposits (Aβ), hyperphosphorylated protein tau, SP-G-containing dots, and GFAP, hippocampal sections from 16 month-old 3xTg-AD mice were immunohistochemically analyzed. Thereby, Figure 8A displayed clustered SP-G-ir dots in the stratum oriens of the CA3 region. Concomitantly, enhanced GFAP-ir (Figure 8A’) was found in the pyramidal layer of the CA3 region, also containing 4G8-immunopositive Aβ plaques (Figure 8A’’). In the overlay (Figure 8A’’’), such blue deposits were encircled by apparently activated astroglia, but were not located near SP-G dots. In the same animal, further triple immunofluorescence labeling displayed SP-G, phospho-tau_202+205_, and GFAP (Data not shown). Figure 8B–B’’’ illustrate a control experiment with switched fluorophores, e.g., green-fluorescent Cy2 visualizing phospho-tau, and blue appearing Cy5-marked astroglia. Such controls verified the unchanged staining patterns for all markers: most SP-G dots appeared as clusters in the polymorphic layer of the dentate gyrus, as well as in the granular cell layer, with a band of neurons filled with hyperphosphorylated tau (Figure 8B’), but which showed less GFAP staining (Figure 8B’’) than in the stratum lacunosum-moleculare of the CA1 field. The overlay (Figure 8B’’’) did not reveal any co-occurrence of the visualized markers, except for one cluster of SP-G-dots in the vicinity of granular cells containing phospho-tau (arrow).

### 3.6. SP-G and Reelin-Containing Dots in the Hippocampus

For the semiquantitative analyses of the clustered dots, hippocampal sections from all animals were double-immunolabeled for SP-G- and Reelin, and the vasculature was counterstained with STL. The apparently age-dependent occurrence of dots was exemplified by comparison of the staining patterns in WT mice aged 6 months (Figure 9A–A’’’) and 24 months (Figure 9B–B’’’). The rostral hippocampus of the younger animal displayed SP-G-ir (Figure 9A) only in the stratum lacunosum-moleculare of the CA1-region. Here, the SP-G-ir dots were frequently not clustered. In parallel, Reelin immunolabeling visualized obviously more numerous dot-like structures (Figure 9A’) in the aforementioned stratum lacunosum-moleculare. Concomitant STL staining (Figure 9A’’) also detected larger vessels in the fissura hippocampalis. Merged staining patterns at a higher magnification (Figure 9A’’’) revealed dot-like structures labeled by Reelin only (green) or containing both SP-G and Reelin (yellow). A representative image of the caudal hippocampus from a 24 month-old WT mouse elucidated clustered dots in the strata radiata of the CA3 and CA1 regions immunopositive for SP-G (Figure 9B) and Reelin (Figure 9B’), separated by the STL-stained fissura hippocampalis. The magnified overlay shows mostly yellowish-appearing dots containing both SP-G and Reelin. Notably, clusters of dots containing SP-G-ir and Reelin-ir were also observed in the piriform cortex (data not shown).

### 3.7. Technical Considerations

Notably, artifact staining by Cy2-donkey-anti-mouse IgG remained restricted to a few poorly perfused animals with stainable amounts of vessel-associated endogenous IgG. The omission of primary antibodies and lectins led to the expected absence of any cellular staining. In further controls focused on age-dependent AD-like alterations, the switch of the fluorophores Cy2 and Cy5, revealing Aβ or phospho-tau and astroglial GFAP, resulted in unchanged staining patterns for all markers. Furthermore, 16 month-old WT mice as well as 3 month-old 3xTg-AD and WT-mice were expectedly devoid of such lesions.

### 3.8. Computer-Based Semiquantification of SP-G-Containing Dots

The aims of the semi-quantification were to determine if there was a statistically significant increase of SP-G-ir with increasing age in 3xTg and WT mice, and to compare the level of SP-G-ir of 3xTg mice with age-matched controls. This analysis included mice in the age groups 3, 6, 12, 16, and ca. 24 months. For each hemisphere, the area of the hippocampus and the combined area of all immunoreactive dots within it was calculated as described above and were displayed in relation to each other. In this way dot-to-tissue-ratios (DTR) were obtained to be compared, with consideration of the animals’ age and genotype.

In WT mice (Figure 10A) no significant differences in DTR could be observed when comparing the age groups of 6, 12, and 16 months (*p* = 1.000; Kruskal-Wallis Test). Therefore, these three groups were summarized as “middle-aged mice”, representing one group to be compared with 3 and 24 month-old mice. This comparison showed a significant increase of DTR in aging WT mice *p* < 0.001 for both group comparisons (Kruskal-Wallis Test). In the group of 3 month-old WT mice, the median share of SP-G-immunopositive dots was 0.000605% of the total hippocampal area (Interquartile range IQR = 0.003234). This shared area increased by the factor 40 to a median 0.024539% (IQR = 0.03) in the ca. 24 month-old animals.

A comparison of the 3xTg mice (Figure 10B) showed no significant differences between the 6 and 12 month-old animals (*p* = 0.125; Kruskal-Wallis Test), as well as between the 16 and ca. 24 month-old animals (*p* = 0.438; Kruskal-Wallis Test). Apart from these two comparisons, there was a significant difference between all age groups (each *p* < 0.001; Kruskal-Wallis Test). The increase in DTR during aging was much larger than in the group of WT animals. In the 3 month-old 3xTg mice, a median of 0.000788% (IQR = 0.001981) of the hippocampal area was SP-G-immunoreactive, increasing by a factor of 124 to a median of 0.097713% (IQR = 0.266297) in the group of ca. 24 month-old mice. Notably, one 25 month-old animal showed an immense DTR of 2.14%.

In comparison to the 3xTg mice, the WT animals apparently displayed a slightly higher DTR in the age groups of 3 and 12 months. However, this difference turned out to be statistically insignificant (3 month-old: *p* = 0.897; 12 month-old: *p* = 0.971; Mann-Whitney-U Test).

In the age groups of 6, 16, and ca. 24 months, the comparison of the two genotypes (Figure 11) showed significantly increased DTRs in the 3xTg mice (*p* < 0.001 in the Mann-Whitney-U Test). Remarkably, some 6 month-old 3xTg animals exhibited DTRs higher than the median DTR of 24 month-old WT-mice. These “outliers” were clearly isolated cases, but such exceptions were not detected within the control group of WT animals. As mentioned above, the DTR increased over the course of aging in both 3xTg and WT mice, but this increase is much lower in the latter ones. When comparing the 6 month-old animals, the median DTR was 2.07 times larger in the 3xTg mice than in the WT group (0.0129% vs. 0.0062%). This factor increased to 7.66 when comparing 16 month-old 3xTg mice with their age-matched control group (0.0520% vs. 0.0068%). The median DTR of ca. 24 month-old 3xTg-mice was 3.98 times greater than the DTR of WT animals of the same age group (0.0977% vs. 0.0245%).

Taken together, a significant increase of SP-G-ir in the hippocampi of both 3xTg- and WT-mice could be observed over the course of aging. Furthermore, this increase was much greater in transgenic animals than in the control group. Therefore, especially the older 3xTg-AD-mice showed significantly elevated DTRs when compared with their age-matched WT controls.

### 3.9. Compartment-Specific Biochemical Analyses of SP-G and Reelin

For the biochemical quantification of SP-G and Reelin by ELISAs, the brains of thirty 3xTg-mice, aged 3 to 16 months, and 27 WT-mice, aged 3 to 24 months, were removed from the skull and prepared as described under Section 2 to obtain the intracellular (ICF), extracellular (ECF), and total protein fraction (TCF).

First, the total protein concentration was measured for all three fractions using a BCA Protein Assay Kit. Next, the concentration of SP-G and Reelin was determined by performing ELISAs on all three fractions according to the manufacturer’s manuals. The specific content of SP-G and Reelin was then compared with the total protein content to generate normalized values and to increase comparability.

Independent of age and genotype, no relevant content of SP-G was detectable in any ECF sample. Therefore, TCF largely corresponds to the intracellular protein fraction and shall be set aside in this investigation. In the following, only the ICF will be considered.

No significant differences were observed when comparing the intracellular SP-G content of WT mice over the course of age (Figure 12A).

The comparison of aging 3xTg-mice, however, shows a clear increase of intracellular SP-G content (Figure 12B). The Tukey post-hoc analysis revealed a significant difference between the 3 and 12 month-old 3xTg animals (*p* = 0.005), as well as between the 3 and 16 month-old mice (*p* < 0.001). Furthermore, both 6 and 12 month-old animals showed a significantly lower intracellular SP-G content than the 16 month-old 3xTg-mice in the Tukey post-hoc analysis (6 vs.16 months: *p* = 0.001; 12 vs. 16 months: *p*= 0.025). On the other hand, no significant differences were observed between 3 and 6 month-old mice (*p* = 0.156), as well as between 6- and 12-month-old mice (*p* = 0.528).

Comparing 3 month-old 3xTg mice with their age-matched controls showed no significant difference in intracellular SP-G content (ANOVA: *p* = 0.924). For the 6, 12, and 16 month-old animals, ANOVA analysis elucidated statistically significant greater contents of intracellular SP-G in 3xTg mice than in age-matched WT mice (6 months: *p* = 0.023; 12 months: *p* = 0.003; 16 months: *p* = 0.001; Figure 12C).

For the 3xTg mice, a strong and statistically significant positive correlation of age and intracellular SP-G content could be determined (Spearman’s ρ = 0.831, *p* < 0.001; Figure 13). This correlation could not be observed in the group of WT animals (Spearman’s ρ = 0.036, *p* = 0.881).

Next, the Reelin levels in the different compartments were analyzed. For the intracellular Reelin content a significant difference was revealed between the differently aged groups of 3xTg mice (ANOVA *p* = 0.023, Figure 14A). The detailed comparison of the age groups showed significant differences for intracellular Reelin between the 3 and 6 month-old (Bonferroni-corrected post hoc test: *p* = 0.049), as well as between the 6 and 16 month-old 3xTg animals (Bonferroni-corrected post hoc test: *p* = 0.049), whereas the other comparisons showed no significance. Remarkably, the intracellular Reelin content appeared to be identical in 3 and 16 month-old animals (Bonferroni-corrected post hoc test: *p* = 1.000). In the extracellular as well as the total protein fraction no significant difference could be observed between the different age groups (*p* = 0.847 and *p* = 0.164).

Furthermore, a significant positive correlation between Reelin content in the TCF and age could be observed in 3xTg mice (Pearson’s Coefficient: r = 0.424, *p* = 0.022; Figure 14B). However, neither the extracellular nor the intracellular Reelin content showed a significant correlation with age (*p* = 0.068 and *p* = 0.130).

The comparison between WT animals of different ages revealed no significant differences (ANOVA: ECF: *p* = 0.659; ICF: *p* = 0.366; TCF: *p* = 0.429). Nevertheless, an inverse correlation between Reelin-content and age was detected in the WT mice. This correlation was pronounced but insignificant in the TCF (R = −0.351, *p* = 0.072) and even stronger and significant in the ICF (R = −0.444, *p* = 0.020; Figure 14C). Thus, both in the 3xTg animals and the comparison group were found to be opposite correlations of Reelin and age.

The comparison of the Reelin content in the brains of age-matched 3xTg and WT animals elucidated significantly elevated Reelin levels in 3xTg-mice in all three compartments and all age groups (ANOVA: *p* = 0.033 *p* < 0.001). The only exception was observed in the ECF of the 3 months-old animals where no significant difference between 3xTg- and WT mice was found (ANOVA: *p* = 0.164). Figure 14D shows data on the group with the oldest animals and serves as an example of the differences in Reelin content between 3xTg and WT mice.

## 4. Discussion

So far, data on SP-G in general are scarce and basically limited to preliminary knowledge on its expression in the lungs and the CNS of rats. Functionally, we are only beginning to understand the potential roles of this novel multifunctional surfactant protein, which seemingly is involved in the rheology of the cerebrospinal fluid and responds to inflammatory stimuli [8,9,10,11,12]. Recently, in situ hybridization (ISH) of mouse sagittal sections revealed SFTA2 mRNA, for instance in the bulbus olfactorius and in the cerebellum [31], which were not addressed in the present study, as well as in the choroid plexus. In order to extend the current understanding of SP-G physiology and regulation under pathological conditions, SP-G expression in the forebrain of mice and SP-G’s relation to the neurovascular unit were elucidated, complemented by a quantitative analysis of SP-G comparing animals of a well-established murine model of AD and WT mice of different age groups.

### 4.1. SP-G-Immunoreactivity within the Neurovascular Unit

Previous work concerning the distribution patterns of SP-G-ir in the brain have remained restricted to one study [11]. The authors reported SP-G co-occurring with AQP4 along cerebral vessels, with endothelial CD31 in the choroid plexus and with NeuN in a small perinuclear area, as well as in pyramid-shaped cortical neurons. Overall, their investigation focused on the association of SP-G with the blood–brain barrier, the blood–cerebrospinal fluid barrier and its involvement in the regulation of CSF rheology, while neuronal SP-G was an unexpected finding and not further explored. Our present work amplifies these data and is specifically focused on the spatial relationships of SP-G-ir with different components of the neurovascular unit, together with its regional expression patterns in the murine forebrain.

First, confocal-laser-scanning microscopy of sections immunolabeled for SP-G and counterstained by the pan-neuronal marker NeuN verified the neuronal expression of SP-G. In detail, the habenula and the hypothalamus were shown to contain numerous neurons co-expressing both proteins. On the contrary, despite a robust fiber staining in the CA2 and CA3 region, hippocampal somata were devoid of clearly detectable SP-G-ir. Thus, SP-G expression is abundant in neurons of certain areas, but SP-G cannot be classified as a general neuronal protein. Surprisingly, in addition to the SP-G-immunoreactive neurons in the hypothalamus and the habenula, the infundibulum displayed neuronal processes with strong SP-G-ir, spanning into the CSF space. In accordance with earlier reports [32], they can be considered as CSF-contacting dendritic processes belonging to bipolar neurons that represent a component of a regulatory circuit connecting the CSF-milieu with secretory neurons in neurohemal areas, for example, the hypothalamus [33,34]. Those bipolar interneurons, first described by Kolmer and Agduhr [35,36,37,38], possess apical extensions spanning into the CSF space with two distinct sensory tasks. Firstly, they monitor CSF composition and secondly, they detect and respond to CSF flow and its alterations [39]. Although our preliminary observation requires further validation, SP-G’s presence in such processes suggests its involvement in the aforementioned circuits, especially when considering the changed CSF-SP-G profile in hydrocephalic conditions and intracranial hemorrhage, as both conditions are associated with altered flow and composition of the CSF [11,12].

Notably, neither oligo- nor astroglia in the analyzed murine forebrains co-expressed SP-G; the only spatial association we found was a group of oligodendroglial cells enwrapping SP-G-expressing neurons in the hypothalamus. More specifically, our study did not confirm the presence of SP-G at the site of the blood–brain barrier, as this was demonstrated earlier in the rat [11]. This raises the question, whether there are differences regarding SP-G expression between mice and rats, or whether the apparent differences are related to experimental conditions.

As mentioned above, we identified few singular Iba-immunopositive cells associated with the choroid plexus that also displayed SP-G-ir. Considering the morphology and location of those Iba-immunoreactive cells, they could correspond to Kolmer cells of monocytic lineage, which contribute to the innate immunity of the CNS and respond to a number of inflammatory stimuli [40,41,42,43]. This preliminary observation may indicate a possible role of SP-G as an immunomodulatory peptide of the CNS and hints to a functional overlap with the collectin-type SPs, which were designated as antibody-equivalent molecules of the innate immune system [44]. However, this co-occurrence could not be reproduced in a further serial staining. Therefore, current evidence is not sufficient in this regard and further research is warranted.

### 4.2. 3xTg Mice as Animal Model with Multifaceted Age-Dependent Deposits

The second aspect of this study, elucidating changes of SP-G under pathological conditions, was based on 3xTg mice mimicking aspects of AD, which were introduced by LaFerla and coworkers [14], and age-matched WT mice without transgenes, but otherwise representing the same genetic background. In 3xTg mice, the inserted mutated human transgenes for presenilin-1, APP1, and tau (e.g., PS1M146V, APPSwedish mutation, and tauP301L) cause synaptic deficits, including disturbed long-term potentiation prior to Aβ deposits and hyperphosphorylated tau, for instance, in the hippocampus [45]. While Aβ deposits are primarily observed in the neocortex and progress to the hippocampus, tau pathology is first apparent in the hippocampus and then spread to the cortex [46]. Notably, intraneuronal accumulation of Aβ precedes extracellular Aβ deposition [47], which is in line with reports of early intracellular Aβx-_42_ in the human brain [48] and in APP/PS1 mice [49], and is considered as a predictor for synaptic dysfunction frequently caused by Aβ oligomers [23,50]. Mastrangelo and Bowers confirmed the predominant restriction of Aβ plaques and hyperphosphorylated tau to the hippocampus, amygdala, and cerebral cortex—the main foci of AD neuropathology in humans—in a comprehensive study [46]. Next, the light and electron microscopic analysis of AD-like pathology in 3xTg mice included the detection of epitopes for abnormally altered tau, and revealed intraneuronal immunoreactivity, for instance, for Alz50, MC1, AT8, and PHF-1 at already 3 weeks of age, whereas straight filaments appeared only in 23 month-old female mice [51]. Based on breeding pairs kindly provided by Drs. LaFerla and Oddo, in 2005, animals from our own breeding elucidated persistently detectable AD-like lesions in aged 3xTg mice and revealed plaques with differently N- and C-terminally truncated Aβ species in 3xTg mice, hardly differing from deposits in senile macaques, baboons, and human cases [52]. Twelve month-old mice displayed hippocampal Aβ deposits in close vicinity to neurons filled with AT8-immunopositive phospho-tau [53].

### 4.3. SP-G and Further Proteins in Age-Dependent Hippocampal Reelin Deposits

Especially aged 3xTg-AD mice were shown to reliably display hippocampal deposits that were distinct from Aβ and phospho-tau. Such deposits contained, for example, the unphosphorylated form of the Kinesin-1 adapter FEZ1 together with Kinesin-1 and the synaptic markers Piccolo, Munc18, and Bassoon [54]. They also resembled age-associated inclusions in WT and transgenic mice [30] which had been previously characterized by Jucker, Walker, and co-workers as laminin-binding protein-immunoreactive granules that are also detectable by Gomori’s methenamine silver stain [29]. Notably, those clustering dots were strongly immunopositive for Reelin, an extracellular protein functionally responsible for CNS development and maturation, as well as synaptic plasticity, throughout the lifespan [55]. Knuesel and co-workers had firstly described such Reelin-positive granular-like deposits in mice, rats, and primates [28]. Knuesel’s group found a significant increase in the number and size of Aβ plaques, as well as aggravated tau pathology, in the hippocampus of animals obtained by crossbreeding heterozygous Reelin knock-out mice (reeler) with 3xTg mice. Kocherhans et al. concluded that age-related Reelin aggregation might play an important role in synaptic dysfunction associated with disturbed Aβ metabolism that is sufficient to enhance tau phosphorylation and to trigger tangle formation [18]. Largely based on immunoelectron microscopy, Doehner et al. detected numerous Reelin deposits with neuritic swellings, containing mitochondria, vacuoles, and cellular debris in naïve mice, which was even more pronounced in prenatally viral-like infected mice [56]. The authors suggested that the dots as neuronal extrusions might be part of a neuroprotective approach to combat age-dependent impaired degradation processes. Notably, Notter and Knuesel reported that Reelin-ir can also be found in neuritic varicosities in the human hippocampal formation of AD patients and non-demented subjects [57].

Our study, for the first time, demonstrated SP-G-ir of most of the afore-mentioned characteristic dot-like Reelin-ir deposits, with a clear age-dependency and a significantly greater amount of SP-G in 3xTg mice than in WT animals. However, as SP-G is clearly not spatially associated with Aβ or phospho-tau but shows a colocalization with Reelin-immunopositive dots and comparable alterations regarding its concentrations, SP-G is apparently linked to this specific histomorphological entity. In line with interpretations provided by Doehner et al. [56], SP-G could therefore be part of a distinct neuroprotective approach against neuronal and synaptic degeneration. It also may be associated with, or a result of, neuroinflammation, which has been postulated as a key player in AD [58]. Nevertheless, the true significance of this can only be hypothesized, because truly functional data to support these considerations are lacking and additional research, for example, involving human samples of AD patients, are required.

### 4.4. Possible Role of SP-G in the (G)lymphatic System—Outlook

An important role for CSF rheology, as well as for brain waste clearance, has been postulated for both, SP-C [7] and SP-G [11]. AD is characterized by an impaired clearance of Aβ from the CNS and 3xTg mice as a murine model for AD exhibit distinct changes regarding SP-C and SP-G concentrations, as well as characteristic neuropathological changes in the respective animals. Weller, Carare, and co-workers clearly demonstrated dysfunctional drainage pathways as a result of age dependent vessel stiffening and Aβ-deposition in the vasculature, causing cerebral angiopathy [59,60]. These authors provided comprehensive data on the peri- and para-vascular clearance of the brain [61,62]. Impaired clearance along the para-venous outflow was shown to largely result from the dysfunction of the water channel AQP4 [63], eventually leading to the concept of the glymphatic (glia-lymphatic) system [64,65]. An important, only recently discovered component of the (g)lymphatic system are lymphatic vessels lining the dural sinuses, which absorb CSF from the adjoining subarachnoid space and interstitial fluid and carry macromolecules to the cervical lymph nodes [66,67]. Different clearing systems in the brain were reviewed, for instance, by Tarasoff-Conway et al. [68] and Natale et al. [69]. However, substantial controversy exists regarding these concepts, which has been summarized nicely by Mestre et al. [70]. In light of the different concepts of brain waste clearance and our findings, future studies on SP-G should therefore elucidate its potential involvement in the (g)lymphatic system and exemplarily comprise electron microscopic analyses. Such investigations might be extended to brain tissues from other mammalian species and autoptic human samples.

## 5. Conclusions

SP-G is apparently synthesized by groups of neurons in the hypothalamus, in the infundibulum, and in the habenula. Its localization in sensory processes of bipolar interneurons contacting the CSF milieu indicates its involvement in regulatory circuits controlling the composition and flow of CSF. A localization of SP-G within Kolmer cells associated with the choroid plexus may hint to a immunological function of SP-G, but could not be validated in this study. SP-G is furthermore present in Reelin-immunoreactive, dot-like deposits in the hippocampus, which are most abundant in old 3xTg-AD mice featuring central elements of AD. Besides its physiological significance, our findings thus indicate a distinct role of SP-G in context of AD’s neuropathology, however, further studies are warranted to validate our findings and to elucidate the specific functions of SP-G in greater detail.

## Figures and Tables

**Figure 1 biomolecules-12-00096-f001:**
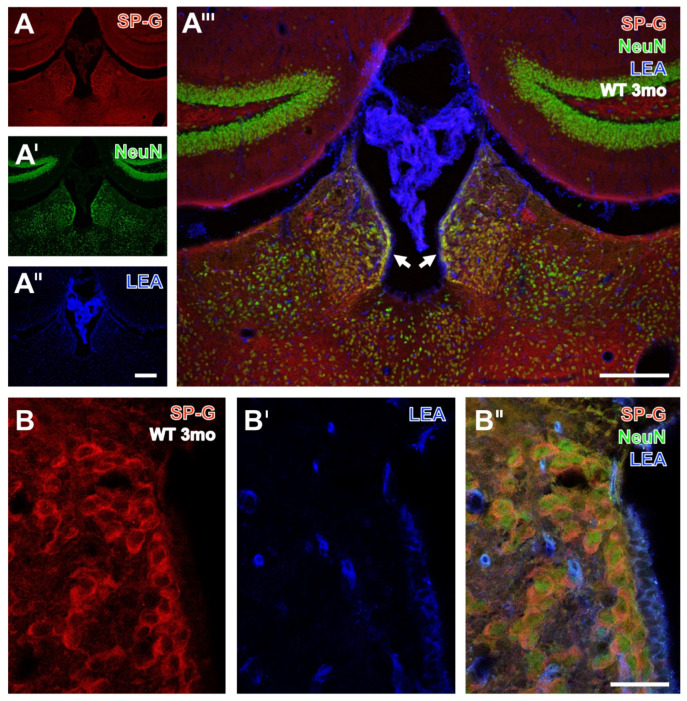
Triple fluorescence labeling of SP-G, LEA-positive vessels, and NeuN in forebrains from 3 month-old WT mice. SP-G-ir (**A**) appears strongest in cells of the habenula, comprises a moderately stained dentate gyrus and a weaker marked granular layer in the hippocampus, and appears mostly lacking in the plexus choroideus. NeuN counterstaining (**A’**) is most prominent in the granular layer with densely packed neuronal somata but indicates also numerous neurons in the habenula and predominantly its lateral part, as well as in the adjacent paraventricular and lateral posterior nuclei of the thalamus. Concomitant labeling with the tomato lectin LEA (**A’’**) reveals vessels in habenula and hippocampus but is much heavier in the plexus choroideus, which is confirmed by the overlay of staining patterns (**A’’’**) that also shows numerous yellowish neurons co-expressing SP-G and NeuN, predominantly in the medial habenula (arrows) and in the paraventricular thalamic nucleus. At a high magnification, confocal laser scanning microscopy reveals SP-G-ir in the rim of cells (**B**) while NeuN-ir remains restricted to the nuclear surroundings (**B’**) and the overlay clearly verifies the co-expression of SP-G and NeuN in several neurons counterstained by the endothelial labeling with LEA in blue. Scale bars: **A’’** (also valid for **A**,**A’**) = 200 µm, **A’’’** = 200 µm, **B’’** (also valid for **B**,**B’**) = 25 µm.

**Figure 2 biomolecules-12-00096-f002:**
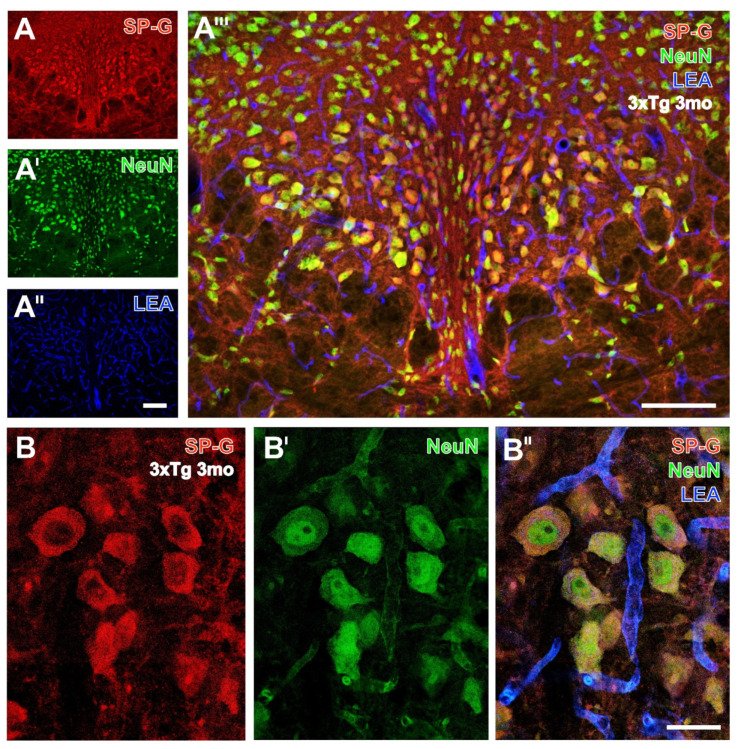
Triple fluorescence labeling of SP-G, LEA-positive vessels, and NeuN in the hypothalamus from a 3 month-old 3xTg mouse. Numerous SP-G-containing neurons of considerably differing sizes are visible in the medial hypothalamus around the posterior hypothalamic nucleus (and around the 3rd ventricle) in (**A**), while NeuN immunolabeling (**A’**) reveals by nuclear staining even more nerve cells. Concomitant detection of LEA-binding sites (**A’’**) shows evenly distributed vessels throughout the presented regions. In the overlay (**A’’’**), a larger portion of neurons displays a green nucleus surrounded by a red SP-immunopositive rim. Notably, some neurons without cut nuclei appear red only. Higher magnified confocal laser scanning micrographs show SP-G-ir predominantly in the rim of nerve cells (**B**) co-expressing NeuN (**B’**), which becomes even clearer in the overlay also showing the LEA-stained vasculature (**B’’**). Scale bars: **A’’** (also valid for **A**,**A’**) = 100 µm, **A’’’** = 100 µm, **B’’** (also valid for **B**,**B’**) = 25 µm.

**Figure 3 biomolecules-12-00096-f003:**
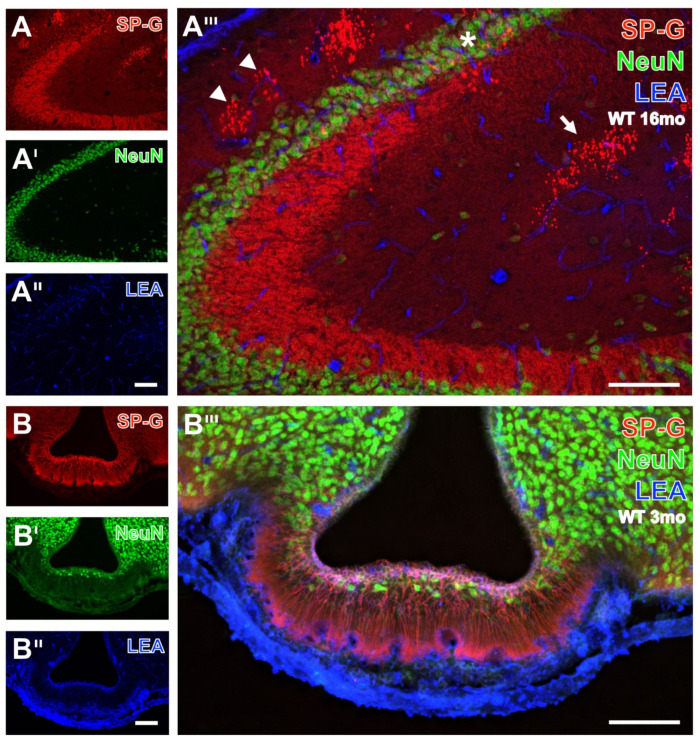
Triple fluorescence labeling of SP-G, LEA-positive vessels, and NeuN in hippocampus (**A**–**A’’’**) and infundibulum (**B**–**B’’’**). At a higher magnification, SP-G-ir in the hippocampus from a 16 month-old WT mouse (**A**) is visible in mossy fibers from the CA3 region and in the CA2 region but appears even stronger in clusters of dots in the stratum lacunosum-moleculare (arrow) and stratum oriens (arrowheads) of the regions CA1 and CA2. The counterstaining of NeuN-ir (**A’**) elucidates densely packed neurons in layers, as well as some single nerve cells in the stratum lacunosum-moleculare of the regions CA2 and CA3, while the vasculature is visualized by LEA-binding sites (**A’’**). The higher-magnified overlay clearly reveals clustered SP-G-immunoreactive dots in the stratum pyramidale of the CA2 region allocated with NeuN-positive neurons (asterisk). Notably, neurons of the granular layer are without detectable SP-G-ir. In (**B**), the infundibulum of a 3 month-old WT mouse shows strong SP-G-ir of fibers from putatively unmyelinated neurons and a somewhat weaker immunosignal of ependymal cells surrounding the lumen of the third ventricle. The latter are intermingled by sparsely distributed NeuN-immunopositive neurons (**B’**), whereas the adjoining arcuate hypothalamic nucleus is filled by densely packed NeuN-stained nerve cells. The strong blue labeling of LEA-binding sites delineating the brain parenchyma is visible already at lower magnification (**B’’**) but is even more clearly shown in the overlay (**B’’’**), where it is dominating in the adjacent ventricle border. Notice the turquoise appearance of tissues bordering the ventricle, indicating the association of SP-G-ir and LEA-binding sites. Scale bars: **B’’** (also valid for **A**–**A’’** and **B**—**B’**) = 100 µm, **B’’’** (also valid for **A’’’**) = 100 µm.

**Figure 4 biomolecules-12-00096-f004:**
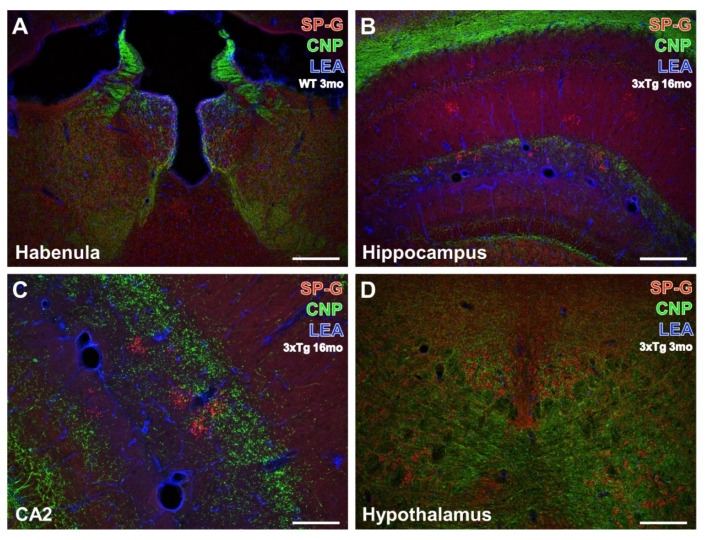
Concomitant fluorescence staining of SP-G, LEA-positive vessels, and oligodendroglial CNP in different forebrain regions. In (**A**), both hemispheres of the habenula from a 3 month-old WT mouse exhibit heavy SP-G labeling that appears strongest in its medial part. More ventrally and medially located intermediodorsal, lateral posterior, parafascicular nuclei, and the posterior complex of the thalamus also display numerous SP-G-containing cells. Further, the immunolabeling of CNP demonstrates many fibrous structures and is most prominent in the striae medullares but is absent in the paraventricular thalamic nucleus. Counterstaining of LEA appears strongest around the ventricle and is moderate in numerous vessels. Note the absence of structures co-expressing SP-G and CNP, as indicated by the lack of the mixed color yellow. The hippocampus of a 16 month-old 3xTg-AD mouse in (**B**) is absent from noticeable cellular SP-G-ir but the neuropil of the gyrus dentatus and strata lacunosum-moleculare and radiatum are faintly stained. In contrast, the strata oriens, lacunosum-moleculare, and radiatum of the CA1 field, as well as the granular layer of the dentate gyrus, contain numerous clusters of SP-G-immunopositive dots. CNP counterstaining reveals massively labeled white matter as well as weaker stained fine processes of the granular layer and coarser structures in the dentate gyrus—intermingled by SP-G-dots and bordered by vessels of the LEA-positive fissura hippocampalis. At a higher magnification, the hippocampal CA2 region from the same old transgenic animal (**C**) shows largely the same spatial relationships between the stained markers as (**B**). Similarly, the lower magnified hypothalamus of a 3 month-old 3xTg-AD mouse (**D**) exhibits SP-G-ir of cells embedded within CNP-immunoreactive processes without co-labeling of both markers in the posterior hypothalamic nucleus. Scale bars: **C** = 100 µm, **D** (also valid for **A**,**B**) = 200 µm.

**Figure 5 biomolecules-12-00096-f005:**
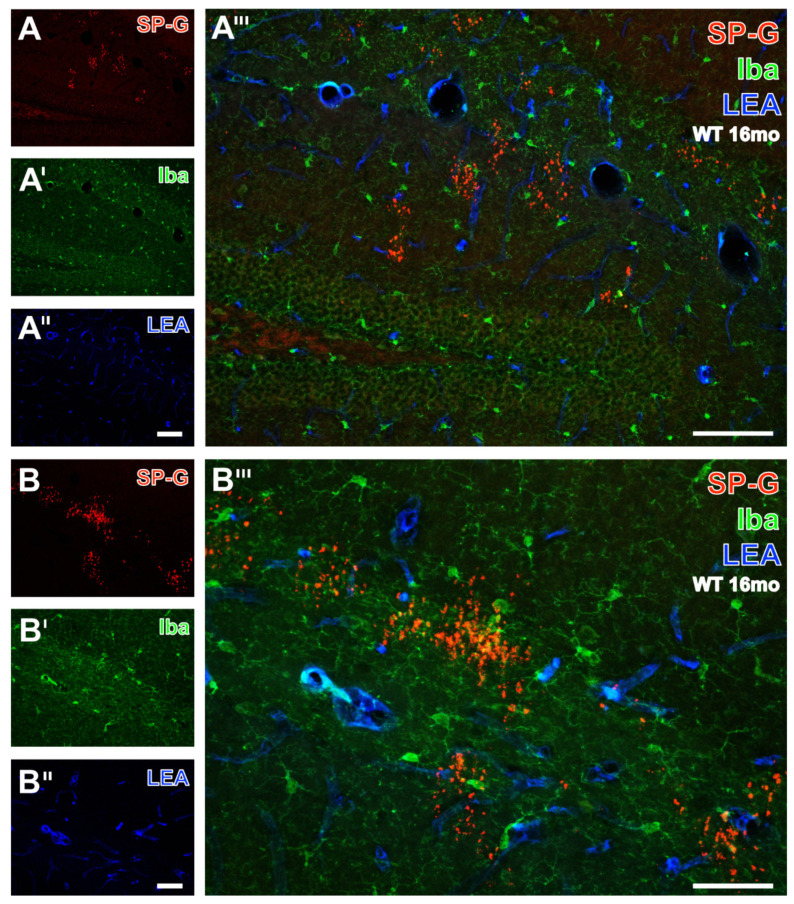
Simultaneous fluorescence labeling of SP-G, LEA-positive vessels, and microglial Iba in the hippocampus of a 16 month-old WT mouse. At a lower magnification, SP-G-ir (**A**) is largely restricted to several clusters of dots in the stratum lacunosum-moleculare of the CA1 field and the molecular layer of the dentate gyrus. In parallel, Iba-immunopositive microglia (**A’**) are evenly distributed in the presented gyrus dentatus and stratum lacunosum-moleculare. Additional vascular LEA-counterstaining (**A’’**) predominantly reveals the fissura hippocampalis, and the overlay (**A’’’**) elucidates several larger perpendicularly cut vessels. Furthermore, the nearly general absence of structures with co-occurring SP-G and Iba becomes obvious. In the same animal, the higher magnified stratum lacunosum-moleculare of the CA2 region reveals SP-G-immunopositive dots singularly and in larger clusters (**B**) surrounded by numerous microglial cells (**B’**) and together with the LEA-stained vasculature (**B’’**). The overlay (**B’’’**) confirms the lacking Iba-ir in dots and displays ramified, but not ameboid, microglia. Scale bars: **A’’** (also valid for **A**,**A’**) = 100 µm, **A’’’** = 100 µm, B’’ (also valid for **B**,**B’**) = 50 µm, **B’’’** = 50 µm.

**Figure 6 biomolecules-12-00096-f006:**
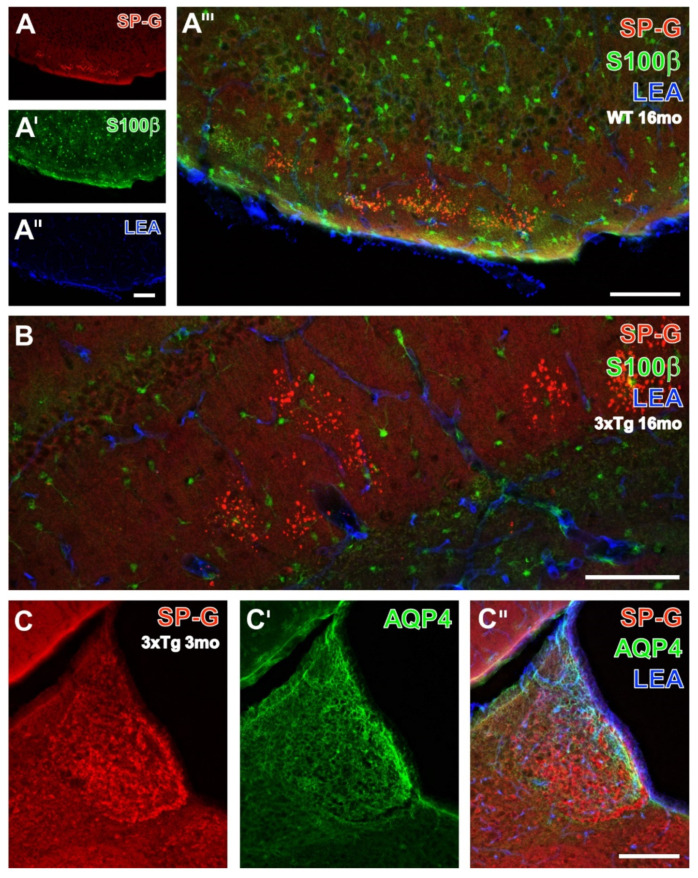
Detection of SP-G-ir and LEA-positive vasculature combined with the immunolabeling of astrocytic S100β protein of a 16 month-old WT mouse (**A**–**A’’’**,**B**) and aquaporin 4 (AQP4) in the habenula of a 3xTg-AD mouse (**C**–**C’’**). SP-G-ir is concentrated in several clusters of numerous dots in the molecular layer of the piriform cortex (**A**), notably without any cellular immunoreactivity for the surfactant protein and counterstained by evenly distributed, S100β-containing astroglia (**A’**) and LEA-stained endothelia (**A’’**). The complete lack of any mixed color clarifies the lacking co-occurrence of the three markers, which is also found in the hippocampal strata lacunosum-moleculare and radiatum, as well as in the stratum moleculare of the dentate gyrus (**B**) displaying SP-G-containing clusters of dots. In contrast, the habenula of a young 3xTg-AD mouse (**C**) shows numerous immunopositive somata, whereas AQP4-ir becomes visible in some vessels, especially in the outer rim of the medial habenula (**C’**). The overlay reveals the absent co-expression of SP-G and AQP4, as well as the predominantly vascular association of AQP-4-ir (green) and LEA-binding sites (blue) that results in turquoise-appearing structures. Scale bars: **A’’** (also valid for **A**,**A’**) = 100 µm, **A’’’** = 100 µm, **B** = 100 µm, **C’’** (also valid for **C**,**C’**) = 200 µm.

**Figure 7 biomolecules-12-00096-f007:**
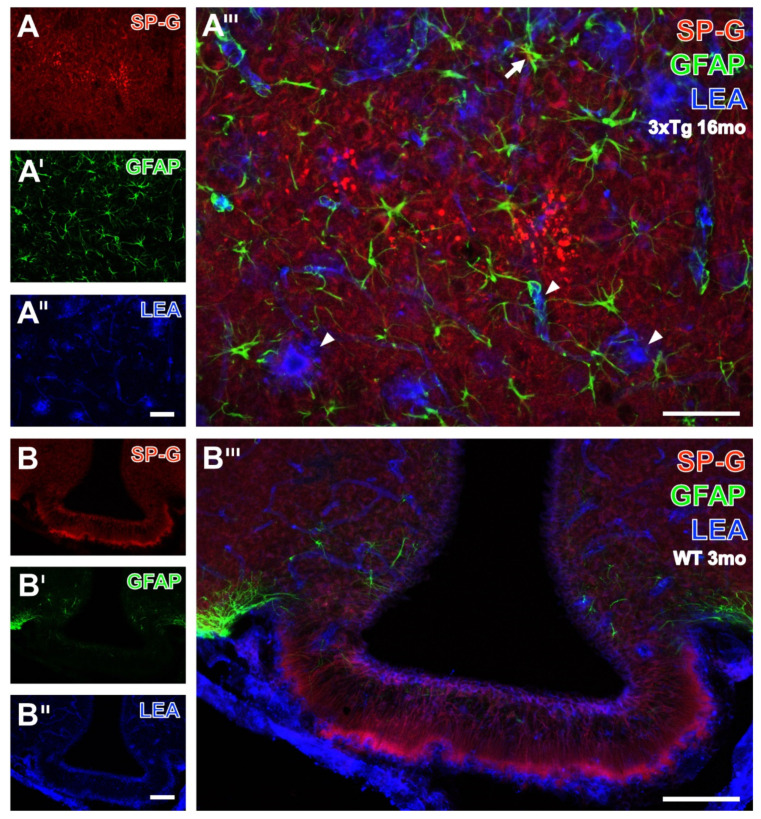
Visualization of SP-G-ir and LEA-binding sites in combination with the immunodetection of astroglial GFAP. As shown at higher magnification (**A**–**A’’’**), the hippocampal CA1 region from a 16 month-old 3xTg mouse displays a faint SP-G-ir of the neuropil and numerous strongly labeled, partially clustered dots (**A**). Counterstaining reveals astrocytes with frequently elaborated processes (**A’**) and LEA-binding of vessels and plaque-like structures (**A’’**). The higher-magnified overlay confirms the occurrence of LEA-positive plaques (arrowheads), surrounded by apparently activated astroglia, as exemplified by an arrow. Additionally, several astrocytic processes targeting the vasculature are visible but distant from the SP-G-containing dots. The same triple staining of the infundibulum from a 3 month-old WT mouse (**B**–**B’’’**) elucidates in (**B**) that numerous SP-G-immunoreactive fibers arise in zone I and stretch through zones II to IV up to the adjacent capillaries. Concomitant GFAP immunolabeling (**B’**) shows moderately stained astrocytes only in the adjacent arcuate hypothalamic nucleus, containing several vessels visualized by LEA (**B’’**), which also stains the ependymal cells encircling the ventricle. In the overlay (**B’’’**), the ependymal layer encircling the ventricle appears mostly purple, indicating the close vicinity of SP-G-ir (red) and LEA-binding sites (blue). Scale bars: **A’’** (also valid for **A**,**A’**) = 50 µm, **A’’’** = 50 µm, **B’’** (also valid for **B**,**B’**) = 100 µm, **B’’’** = 100 µm.

**Figure 8 biomolecules-12-00096-f008:**
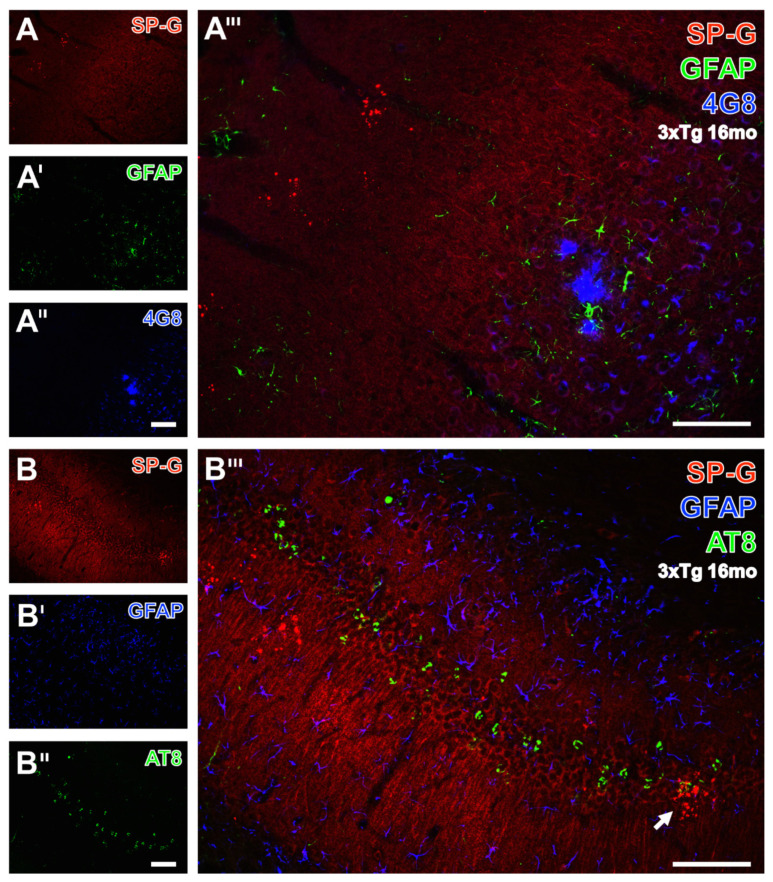
Triple fluorescence labeling of SP-G, astroglia, and β-amyloid deposits (Aβ) or hyperphosphorylated protein tau (phospho-tau) in the hippocampus of a 16 month-old 3xTg-AD mouse. In (**A**), two clusters of SP-G-containing red fluorescent dots in the stratum oriens of the CA3 region become visible. Concomitantly, the GFAP staining (**A’**) is not evenly distributed and appears strongest in the pyramidal layer of the CA3 region. The same region displays three 4G8-positive, Aβ-containing plaques (**A’’**) located in close vicinity. These blue deposits are surrounded by activated astrocytes, whereas plaques and astroglia are seen only distant from SP-G dots. (**B**–**B’’’**) demonstrate the simultaneous staining of SP-G, phospho-tau, and GFAP, exemplifying the control experiment with switched fluorophores, e.g., green-fluorescent Cy2 for the AD-like lesions and Cy5 (color-coded in blue) for astroglia. Dot-like SP-G-ir (**B**) appears mostly clustered in the polymorph layer of the dentate gyrus and in the granular cell layer, which contains numerous neurons filled with hyperphosphorylated tau (**B’’**) but displays (in **B’**) weaker GFAP-ir than, for instance, the stratum lacunosum-moleculare of the CA1 field. Merging of staining patterns (**B’’’**) does not reveal co-localization of the visualized markers, except for one cluster of SP-G-dots in the neighborhood of neuronal phospho-tau in the granular layer (arrow). Scale bars: **B’’** (also valid for **A**–**A’’**,**B**,**B’**) = 100 µm, **B’’’** (also valid for **A’’’**) = 100 µm.

**Figure 9 biomolecules-12-00096-f009:**
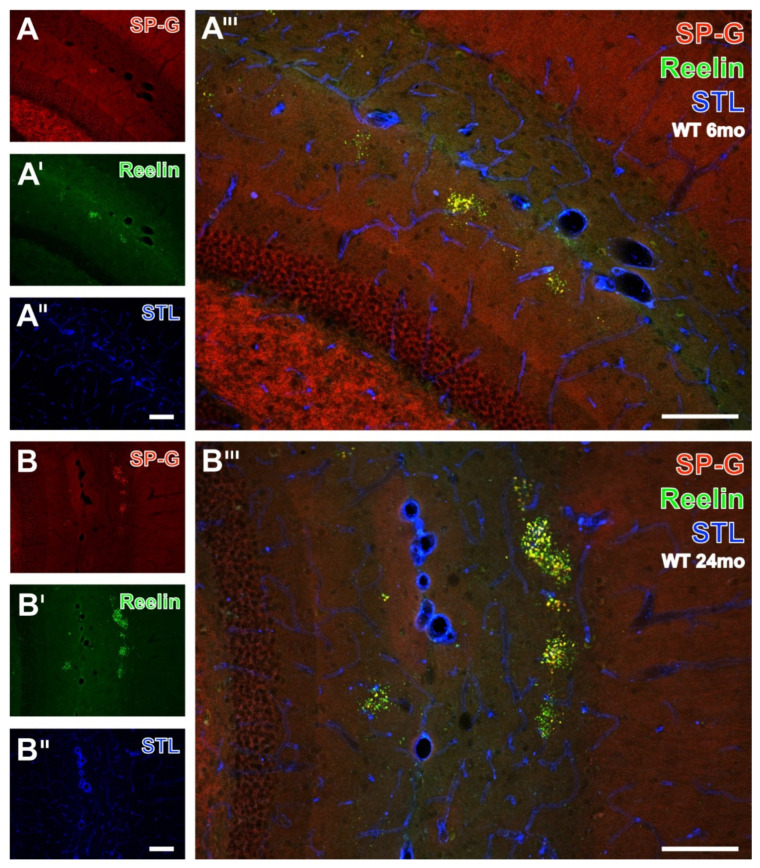
Double immunolabeling of SP-G- and Reelin combined with the detection of endothelia by STL in the hippocampus of 6 and 24 month-old WT mice. In a 6 month-old animal, the presented SP-G-ir remains restricted to the stratum lacunosum-moleculare of the CA1-region in a more rostrally located part of the hippocampus and is in some cases not clustered (**A**). Concomitantly, Reelin-ir reveals apparently more dot-like structures (**A’**) exclusively in the stratum lacunosum-moleculare. Additional counterstaining of endothelia with STL (**A’’**) also visualizes some perpendicularly cut vessels of the fissura hippocampalis. The overlay (**A’’’**) indicates the lacking co-occurrence of vasculature and dot-like structures which are either mono-labeled by Reelin (green) or contain SP-G as well as Reelin (yellow). The same triple staining in the caudal, laterally positioned hippocampus from a 24 month-old WT mouse detects, in (**B**), four clusters, with SP-G-ir in the stratum radiatum of the CA2 region and one cluster in the stratum moleculare of the dentate gyrus, which are found to be also Reelin-immunopositive in (**B’**). Simultaneously, vascular STL-binding sites (**B’’**) clearly demarcate the fissura hippocampalis between the clusters of dots in the hippocampal stratum radiatum and the stratum moleculare of the dentate gyrus. The overlay (**B’’’**) shows numerous clusters predominantly containing yellowish-appearing dots, which are double-positive for SP-G and Reelin, some green dots without at least moderate SP-G-ir, and a few orange dots exhibiting weak Reelin-ir. Scale bars: **B’’** (also valid for **A**–**A’’**,**B**,**B’**) = 100 µm, **B’’’** (also valid for **A’’’**) = 100 µm.

**Figure 10 biomolecules-12-00096-f010:**
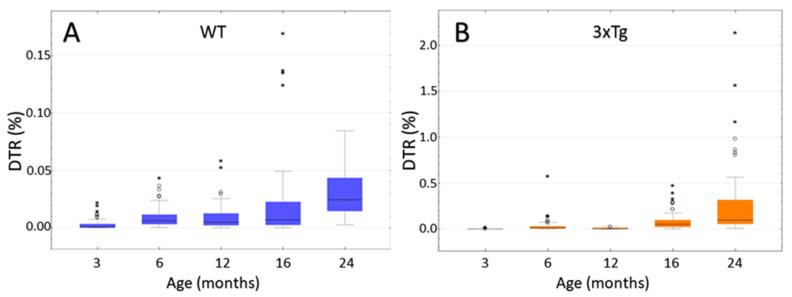
Age-dependency of hippocampal SP-G-immunopositive dot-to-tissue-ratios (DTR given as percentage). (**A**) WT mice: Since there is no significant difference between the 6, 12, and 16 month-old animals, these three groups were summarized as “middle-aged mice” and compared to the other two age groups. A significant difference in DTR could be shown between the young (3 months), middle-aged, and old WT-mice (*p* < 0.001 for both group comparisons; Kruskal-Wallis Test) with a clear increase over the course of aging. (**B**) 3xTg mice: No significant differences in DTR could be shown between the 6 and 12 month-old animals (*p* = 0.125; Kruskal-Wallis Test) and the 16 and ca. 24 month-old animals (*p* = 0.438; Kruskal-Wallis Test). Comparisons of all the other age groups showed highly significant differences (each *p* < 0.001; Kruskal-Wallis Test) with a clear increase of DTR during aging. Remarkably, in one old animal, 2.14% of the hippocampal area was occupied by SP-G-mmunoreactive dots. Notice that the ordinate is scaled differently in both graphs. ◦: Indicates data values lying 1.5 to 3× of the IQR above the mean value. *: Indicates data values lying >3× of the IQR above the mean value.

**Figure 11 biomolecules-12-00096-f011:**
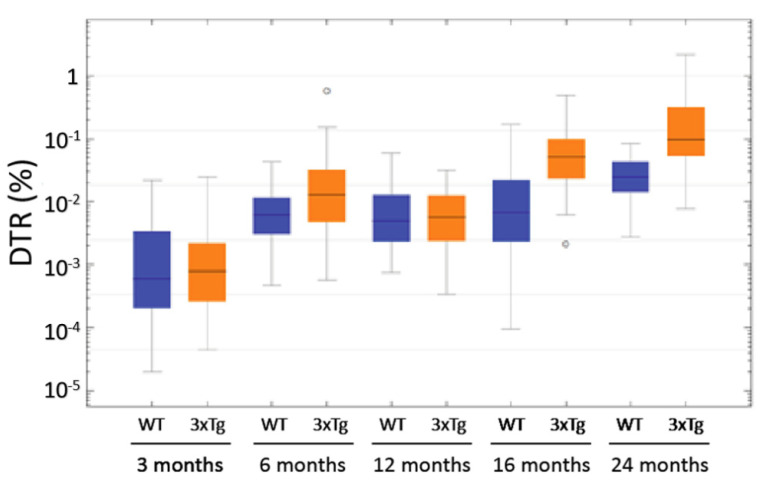
Logarithmically scaled box plot of the SP-G-immunopositive dot-to-tissue ratios (DTR) as a function of age and genotype. In the age groups of 3 and 12 month-old animals, there were no significant differences between 3xTg and WT mice (3 month-old: *p* = 0.897; 12 months-old: *p* = 0.971; Mann-Whitney-U Test). In the other age groups, a significant increase of the DTR in 3xTg mice when compared to their age-matched controls was evident (each *p* < 0.001 in the Mann-Whitney-U Test). Furthermore, this figure shows the increase in DTR over the course of ageing in both 3xTg- and WT-mice. ◦: Indicates data values lying 1.5 to 3× of the IQR away from the mean value.

**Figure 12 biomolecules-12-00096-f012:**
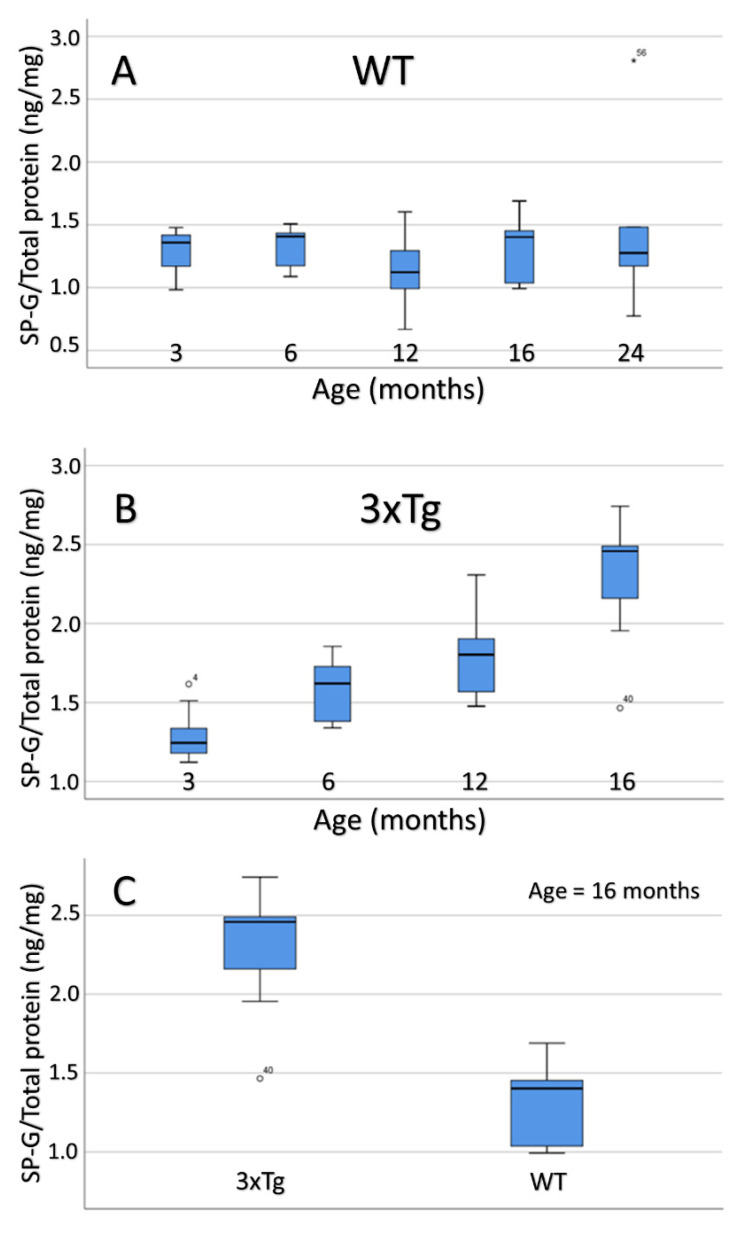
Box plot representation of the age-dependent SP-G content within the total intracellular brain protein fraction (specified in ng/mg). (**A**) WT mice: There are no statistically significant differences between the age groups. (**B**) In 3xTg mice, a clear increase of the SP-G content during aging becomes evident. The Tukey post-hoc analysis detected significant differences between all age groups except for the comparisons between 3 and 6 month-old animals (*p* = 0.156) and 6 and 12 month-old mice (*p* = 0.528). (**C**) Box plot representation of the SP-G content within the total intracellular protein fraction of 16 month-old mice brain depending on genotype (specified in ng/mg). ANOVA analysis revealed a significantly increased SP-G content in 3xTg mice when compared to their age-matched WT controls (*p* = 0.001). ◦: Indicates data values lying 1.5 to 3× of the IQR away from the mean value. *: Indicates data values lying >3× of the IQR away from the mean value.

**Figure 13 biomolecules-12-00096-f013:**
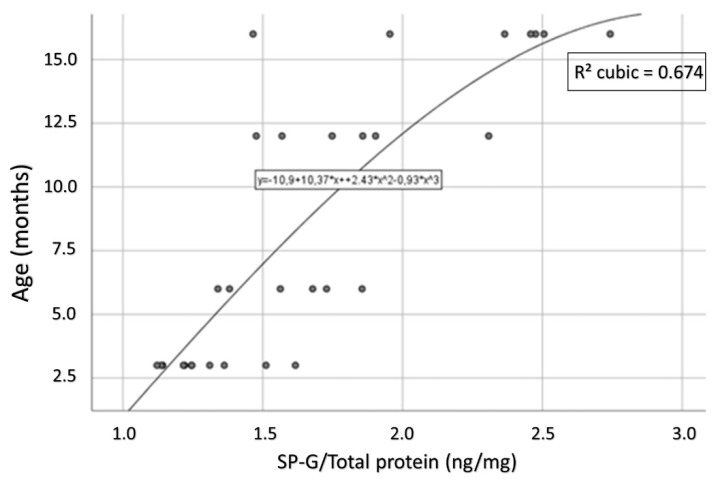
Representation of the correlation between age and intracellular SP-G content in 3xTg mice. This correlation is strong and statistically significant (Spearman’s ρ = 0.831, *p* < 0.001).

**Figure 14 biomolecules-12-00096-f014:**
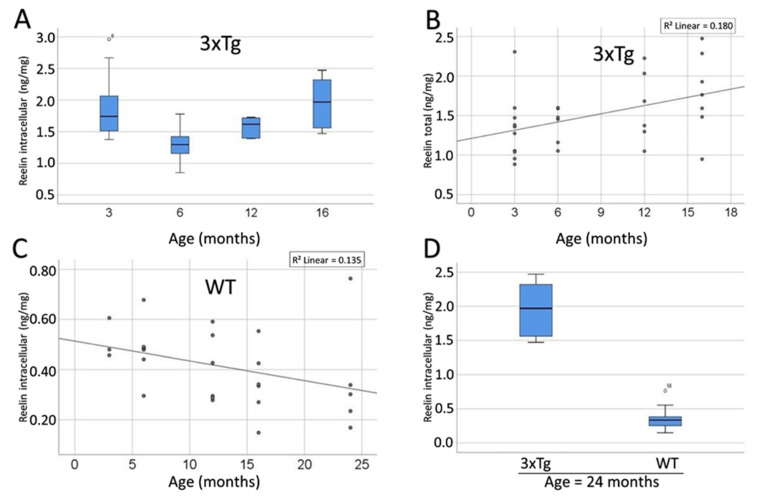
The concentration of Reelin significantly rises over the course of ageing in 3xTg mice, whereas WT animals show a significant decline. (**A**) Boxplot representation of the age-dependent Reelin content within the intracellular fraction of the brains of 3xTg mice (specified in ng/mg). Overall, a significant difference between the age groups was observed (ANOVA *p* = 0.023). The detailed comparison of the single groups revealed significant differences between the 3 and 6 month-old animals (Bonferroni-corrected post hoc test: *p* = 0.049) and between the 6 and 16 month-old mice (Bonferroni-corrected post hoc test: *p* = 0.049). (**B**) Correlation between age and Reelin content in the TCF in 3xTg mice. This correlation is positive and statistically significant (R = 0.424, *p* = 0.022). (**C**) Correlation between age and Reelin content in the ICF in WT mice. This correlation is negative and statistically significant (Pearson’s Coefficient: r = −0.444, *p* = 0.020). (**D**) Representation of the intracellular Reelin content of ≥16 month-old mice in dependency of their genotype. 3xTg mice show significantly elevated levels of intracellular Reelin (ANOVA: *p* = 0.033). ◦: Indicates data values lying 1.5 to 3× of the IQR away from the mean value. *: Indicates data values lying >3× of the IQR away from the mean value.

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
