# Peer review of "Surfactant Protein-G in Wildtype and 3xTg-AD Mice: Localization in the Forebrain, Age-Dependent Hippocampal Dot-like Deposits and Brain Content"

_biomolecules, 2022, doi:10.3390/biom12010096_

Round 1

Reviewer 1 Report

This work by Meinicke et al., exhaustively documents the spatiotemporal presence of surfactant protein G across wild type and 3xTg-AD mice. While the amount of work is impressive, the manuscript would benefit from some changes to the figures and text to improve accessibility and understanding of its importance. 

  1. Figure 2 feels out of place as the 3xTg mouse model is introduced later in Figure 4.
  2. Please label the pictures themselves if the stainings are from WT or transgenic tissues, as well as the age. This makes reading the figures much easier.
  3. More detailed labels for anatomical structures in the pictures would also be good to have.
  4. It is unclear how reelin is connected to the story in the introduction and abstract.
  5. It would be important if the Authors discuss the specificity of the SP-G antibody, especially when analyzing immunoreactive dots. Was a blocking peptide used?
  6. Higher magnification images and antibody specificity checks of the median eminence is required, especially since this region is notorious for fibrous back ground staining.
  7. While hippocampal neurons seem devoid of SP-G, figure 3 shows strong staining in CA2/3. This should be discussed.
  8. Does the co-localization with reelin only occur in the hippocampus (e.g. fig. 6a''')?
  9. Please explain how SP-G accumulation in the hippocampus can contribute to immune modulation, as there is no close proximity of microglia, astrocytes or blood vessels associated with the deposit.
  10. As reelin is associated with AB, and SP-G with reelin in the data presented, can the Authors discuss why SP-G does not co-localize with 4G8 in the hippocampus (fig 8)? Are these puncta reelin-negative and there is a direct interaction with reelin-SP-G only?

Author Response

Dear reviewer 1,

We want to thank you for your constructive criticism. Each of your comments was carefully considered and addressed accordingly.

  1. Figure 2 feels out of place as the 3xTg mouse model is introduced later in Figure 4.

Dear reviewer,

Thank you for pointing out that the presentation of figures is confusing without further explanations. As the order of pictures is apparently not self-explanatory, the rationale behind it is now explained under 3.1. 

  1. Please label the pictures themselves if the stainings are from WT or transgenic tissues, as well as the age. This makes reading the figures much easier.

We would like to thank the reviewer for these important suggestions. Despite the

 limited time for the revision, we improved Figures 1-9 by adding the requested data

 on genetic background (e.g. WT or 3xTg) and age of the animals.

  1. More detailed labels for anatomical structures in the pictures would also be good to have.

Having in mind the already rather long figure legends containing numerous hints to stained structures we prefer to avoid an over-illustration by detailed labels of anatomical structures which would require the explanation of their abbreviations in the figure legends.

  1. It is unclear how reelin is connected to the story in the introduction and abstract.

Dear reviewer, thank you very much for this important comment. Reelin is a predominantly extracellular glycoprotein involved in neuronal maturation and synaptic plasticity. Functional Reelin restricts hyperphosphorylation of tau and the formation of β-amyloid quite upstream in each of the respective pathways. Accumulation of non-functional Reelin precedes the ‚classic‘, spatially distinct neuropathological manifestations of AD, such as β-amyloid and phospho-tau (Kobro-Flatmoen et al., 2016), enhances the latter and is linked to synaptic dysfuntion. Reelin deposits, being distinct from β-amyloid and phospho-tau, have therefore been used as a reference for the progress of AD-like neuropathological changes shown by  this animal model in earlier works, which is why we also implemented them as such. The abstract and introduction were changed accordingly.

  1. It would be important if the Authors discuss the specificity of the SP-G antibody, especially when analyzing immunoreactive dots. Was a blocking peptide used?

We would like to thank for the critical remarks and have clarified sentences regarding anti-SP-G and its specificity.

Due to the fact that a recombinant protein had served as immunogen for the generation of anti-SP-G, a control peptide was not available.

Under 2.3. we specify now by printing “…with an affinity-purified rabbit antibody”. The following sentence was completed by inserting “product number PAD755Hu01” between “USA;” and “;1:300”.

  1. Higher magnification images and antibody specificity checks of the median eminence is required, especially since this region is notorious for fibrous background staining.

In control experiments, the omission of primary antibodies and of the biotinylated tomato lectin LEA resulted in the absence of stained cellular structures – despite the widely accepted fact that the stained type of tissue is susceptible to background staining.          

The currently presented pictures were already acquired with a higher magnification than most other micrographs, which are shown in our manuscript. We feel that the implementation of images with even higher magnifications would adversely affect the desired overview, and additional color plates would cause an over-illustration considering the already great number of figures in our study. However, if you feel an additional image is of great importance, we might provide such an additional figure with some delay.

  1. While hippocampal neurons seem devoid of SP-G, figure 3 shows strong staining in CA2/3. This should be discussed.

This very helpful comment has been addressed in our manuscript by newly mentioning the fiber staining in the CA2 and CA3 region (4.1., second paragraph): “On the contrary, despite a robust fiber staining in the CA2 and CA3 region, hippocampal somata were devoid of clearly detectable SP-G-ir.” The respective legend for Figure 3 was revised, following your suggestion accordingly (starting in the second line ): “SP-G-ir in the hippocampus…. is visible in mossy fibers from the CA3 region and in the CA2 region…”

  1. Does the co-localization with reelin only occur in the hippocampus (e.g. fig. 6a''')?

To answer this question, we would like to add as last sentence under 3.6:

“Notably, clusters of dots containing SP-G-ir and Reelin-ir were also observed in the pi-riform cortex (data not shown).” Please note, that the piriform cortex displaying numerous clusters of SP-G-immunoreactive dots without Reelin counterstaining is shown in Figure 6A.

  1. Please explain how SP-G accumulation in the hippocampus can contribute to immune modulation, as there is no close proximity of microglia, astrocytes or blood vessels associated with the deposit.

Dear reviewer 1, we fully agree with your comment, hippocampal accumulation of SP-G does not allow any interpretation with respect to immunological functions. In our opinion, studies  performed by Mittal et al. (2012) and  Krause et al. (2019) certainly justify to consider SP-G as immunologically relevant molecule, but our study did not substantiate immunological functions of SP-G. As discussed in brief, few images in our investigation indicated an association between Kolmer epiplexus cells – representing dendritic cells of the CNS – and SP-G, but we were not able to reliably reproduce this finding, which is mostly related to technical difficulties associated with fluorescence-immunolabeling of the choroid plexus. Therefore we only briefly mentioned this aspect and suggested to further investigate the immunological roles of SP-G in future investigations.

  1. As Reelin is associated with AB, and SP-G with Reelin in the data presented, can the Authors discuss why SP-G does not co-localize with 4G8 in the hippocampus (fig 8)? Are these puncta Reelin-negative and there is a direct interaction with Reelin-SP-G only?

Dear reviewer 1, in this investigation and earlier studies as well, dot-like Reelin-ir deposits have been described as histopathological entities sometimes overlapping with- and sometimes separate from β-amyloid and phospho-tau. Apparently, Reelin depositions occur in both ways, associated with β-amyloid / hyperphosphorylated tau and distinct from them. This observation is probably related to the fact that the pathological cascades causing β-amyloidosis and tau hyperphosphorylation appear separated from neuropathological alterations causing  reelin- (and SP-containing) dot-like deposits.

Again, we want to thank you for your consideration and helpful comments, which helped to improve our manuscript significantly. We are looking forward to any further question that you may have.

Reviewer 2 Report

Ref. biomolecules-1525387

The authors studied surfactant protein-G (SP-G) in wild-type and 3xTg-AD-mice and observed a significant increase of SP-G in old 3xTg-AD mice as compared to age-matched wild type mice. Furthermore, 3xTg-AD mice displayed SP-G containing dots in the hippocampus separately for age dependent deposits of beta-amyloid and p-tau.

Since the presence of SP-G in the CNS has been recognized only recently and the possible role of this protein in the normal brain, in ageing and in neurodegeneration is currently under investigation, the present paper is very interesting and very welcomed.

The study is well design and conducted and statistics are adequate. I have no comments and I would like to see it published.

Author Response

Dear Ms. Luo, dear Prof. Rossner,

we want to thank you very much for giving us the opportunity to submit a revised version of our manuscript. We want to thank both reviewers for their time and effort, which they invested in our manuscript. We feel the manuscript has benefitted from the review process. Each comment was addressed accordingly. Please find our point to point response as follows:

Dear reviewer 1,

We want to thank you for your constructive criticism. Each of your comments was carefully considered and addressed accordingly.

  1. Figure 2 feels out of place as the 3xTg mouse model is introduced later in Figure 4.

Dear reviewer,

Thank you for pointing out that the presentation of figures is confusing without further explanations. As the order of pictures is apparently not self-explanatory, the rationale behind it is now explained under 3.1. 

  1. Please label the pictures themselves if the stainings are from WT or transgenic tissues, as well as the age. This makes reading the figures much easier.

We would like to thank the reviewer for these important suggestions. Despite the

 limited time for the revision, we improved Figures 1-9 by adding the requested data

 on genetic background (e.g. WT or 3xTg) and age of the animals.

  1. More detailed labels for anatomical structures in the pictures would also be good to have.

Having in mind the already rather long figure legends containing numerous hints to stained structures we prefer to avoid an over-illustration by detailed labels of anatomical structures which would require the explanation of their abbreviations in the figure legends.

  1. It is unclear how reelin is connected to the story in the introduction and abstract.

Dear reviewer, thank you very much for this important comment. Reelin is a predominantly extracellular glycoprotein involved in neuronal maturation and synaptic plasticity. Functional Reelin restricts hyperphosphorylation of tau and the formation of β-amyloid quite upstream in each of the respective pathways. Accumulation of non-functional Reelin precedes the ‚classic‘, spatially distinct neuropathological manifestations of AD, such as β-amyloid and phospho-tau (Kobro-Flatmoen et al., 2016), enhances the latter and is linked to synaptic dysfuntion. Reelin deposits, being distinct from β-amyloid and phospho-tau, have therefore been used as a reference for the progress of AD-like neuropathological changes shown by  this animal model in earlier works, which is why we also implemented them as such. The abstract and introduction were changed accordingly.

  1. It would be important if the Authors discuss the specificity of the SP-G antibody, especially when analyzing immunoreactive dots. Was a blocking peptide used?

We would like to thank for the critical remarks and have clarified sentences regarding anti-SP-G and its specificity.

Due to the fact that a recombinant protein had served as immunogen for the generation of anti-SP-G, a control peptide was not available.

Under 2.3. we specify now by printing “…with an affinity-purified rabbit antibody”. The following sentence was completed by inserting “product number PAD755Hu01” between “USA;” and “;1:300”.

  1. Higher magnification images and antibody specificity checks of the median eminence is required, especially since this region is notorious for fibrous background staining.

In control experiments, the omission of primary antibodies and of the biotinylated tomato lectin LEA resulted in the absence of stained cellular structures – despite the widely accepted fact that the stained type of tissue is susceptible to background staining.          

The currently presented pictures were already acquired with a higher magnification than most other micrographs, which are shown in our manuscript. We feel that the implementation of images with even higher magnifications would adversely affect the desired overview, and additional color plates would cause an over-illustration considering the already great number of figures in our study. However, if you feel an additional image is of great importance, we might provide such an additional figure with some delay.

  1. While hippocampal neurons seem devoid of SP-G, figure 3 shows strong staining in CA2/3. This should be discussed.

This very helpful comment has been addressed in our manuscript by newly mentioning the fiber staining in the CA2 and CA3 region (4.1., second paragraph): “On the contrary, despite a robust fiber staining in the CA2 and CA3 region, hippocampal somata were devoid of clearly detectable SP-G-ir.” The respective legend for Figure 3 was revised, following your suggestion accordingly (starting in the second line ): “SP-G-ir in the hippocampus…. is visible in mossy fibers from the CA3 region and in the CA2 region…”

  1. Does the co-localization with reelin only occur in the hippocampus (e.g. fig. 6a''')?

To answer this question, we would like to add as last sentence under 3.6:

“Notably, clusters of dots containing SP-G-ir and Reelin-ir were also observed in the pi-riform cortex (data not shown).” Please note, that the piriform cortex displaying numerous clusters of SP-G-immunoreactive dots without Reelin counterstaining is shown in Figure 6A.

  1. Please explain how SP-G accumulation in the hippocampus can contribute to immune modulation, as there is no close proximity of microglia, astrocytes or blood vessels associated with the deposit.

Dear reviewer 1, we fully agree with your comment, hippocampal accumulation of SP-G does not allow any interpretation with respect to immunological functions. In our opinion, studies  performed by Mittal et al. (2012) and  Krause et al. (2019) certainly justify to consider SP-G as immunologically relevant molecule, but our study did not substantiate immunological functions of SP-G. As discussed in brief, few images in our investigation indicated an association between Kolmer epiplexus cells – representing dendritic cells of the CNS – and SP-G, but we were not able to reliably reproduce this finding, which is mostly related to technical difficulties associated with fluorescence-immunolabeling of the choroid plexus. Therefore we only briefly mentioned this aspect and suggested to further investigate the immunological roles of SP-G in future investigations.

  1. As Reelin is associated with AB, and SP-G with Reelin in the data presented, can the Authors discuss why SP-G does not co-localize with 4G8 in the hippocampus (fig 8)? Are these puncta Reelin-negative and there is a direct interaction with Reelin-SP-G only?

Dear reviewer 1, in this investigation and earlier studies as well, dot-like Reelin-ir deposits have been described as histopathological entities sometimes overlapping with- and sometimes separate from β-amyloid and phospho-tau. Apparently, Reelin depositions occur in both ways, associated with β-amyloid / hyperphosphorylated tau and distinct from them. This observation is probably related to the fact that the pathological cascades causing β-amyloidosis and tau hyperphosphorylation appear separated from neuropathological alterations causing  reelin- (and SP-containing) dot-like deposits.

Again, we want to thank you for your consideration and helpful comments, which helped to improve our manuscript significantly. We are looking forward to any further question that you may have.

Dear reviewer 2,

We want to express our gratitude for your very kind review. We are happy you find our manuscript interesting and suitable for publication in the special issue of Biomolecules. Again, thank you very much for all the time and effort that you invested.
